# IDOL: Meeting Diverse Distribution Shifts with Prior Physics for Tropical Cyclone Multi-Task Estimation

**Hanting Yan**[1], **Pan Mu**[1], **Shiqi Zhang**[1], **Yuchao Zhu**[1], **Jinglin Zhang**[2], **Cong Bai**[1,3]*

[1]College of Computer Science, Zhejiang University of Technology, China
[2]School of Control Science and Engineering, Shandong University, China
[3]Zhejiang Key Laboratory of Visual Information Intelligent Processing, China

## Abstract

Tropical Cyclone (TC) estimation aims to accurately estimate various TC attributes in real time. However, distribution shifts arising from the complex and dynamic nature of TC environmental fields, such as varying geographical conditions and seasonal changes, present significant challenges to reliable estimation. Most existing methods rely on multi-modal fusion for feature extraction but overlook the intrinsic distribution of feature representations, leading to poor generalization under out-of-distribution (OOD) scenarios. To address this, we propose an effective **I**dentity **D**istribution-**O**riented Physical Invariant **L**earning framework (**IDOL**), which imposes identity-oriented constraints to regulate the feature space under the guidance of prior physical knowledge, thereby dealing distribution variability with physical invariance. Specifically, the proposed IDOL employs the wind field model and dark correlation knowledge of TC to model task-shared and task-specific identity tokens. These tokens capture task dependencies and intrinsic physical invariances of TC, enabling robust estimation of TC wind speed, pressure, inner-core, and outer-core size under distribution shifts. Extensive experiments conducted on multiple datasets and tasks demonstrate the outperformance of the proposed IDOL, verifying that imposing identity-oriented constraints based on prior physical knowledge can effectively mitigates diverse distribution shifts in TC estimation. Code is available at `https://github.com/Zjut-MultimediaPlus/IDOL`.

## 1 Introduction

As a typical natural disaster, Tropical Cyclones (TCs) endanger human life and the environment each year. Accurate estimation of TC intensity and size is thus essential in the field of weather service, as it helps reduce casualties and property losses. Driven by the strong capability of deep learning in feature representation and nonlinear modeling, recent TC multi-task estimation methods [7, 8] primarily employ deep neural networks to jointly estimate multiple TC attributes, including intensity and size. Generally speaking, deep learning models rely on the assumption that the training and test data are drawn from the same underlying distribution, referred to as in-distribution [11], to make reliable predictions. However, in real-world scenarios, the spatiotemporal heterogeneity of TC environmental fields often gives rise to complex and diverse developmental pathways [10], resulting in out-of-distribution (OOD) data during inference. Consequently, ignoring the inherent distribution of network embeddings, existing methods may fail to learn features that capture all possible variations in TC evolution. This, in turn, severely limits the models' ability to generalize effectively when learning from previously unseen TCs. To verify the presence of OOD data, we use violin plots to visualize the distributions of input $X$ and output $Y$ across different datasets or tasks. As shown in Figure 7 in Appendix A, the varying shapes and spreads of the violins reveal notable discrepancies in the TC data distributions.

---

*Corresponding author. Email: congbai@zjut.edu.cn

These observations highlight the importance of improving model generalization by explicitly addressing the OOD issue in TC estimation. Considering the spatiotemporal and multivariate correlations among multiple TC attributes, based on previous studies [12, 13], we attribute the OOD issue in TC estimation to three types of distribution shifts. As shown in Figure 1, covariate shift and label shift refer to changes in the input and output distributions between the seen and unseen datasets, i.e., $P(X)$ vs. $P(X')$ and $P(Y)$ vs. $P(Y')$, respectively. Concept shift denotes variations in the conditional probabilities $P(Y \mid X)$, arising when different TC attributes follow distinct developmental patterns and data distributions. The detailed definitions and underlying causes of these shifts are provided in Appendix A. To cope with various distribution shifts, one important approach is invariant feature learning, which aims to extract features with certain invariance across both training and other unseen domains [14]. From this perspective, two key challenges remain to be addressed. **C1:** What can represent the invariance in a specific domain, i.e., TC multi-task estimation? **C2:** How can this invariance be modeled and utilized to handle the aforementioned three types of distribution shifts?

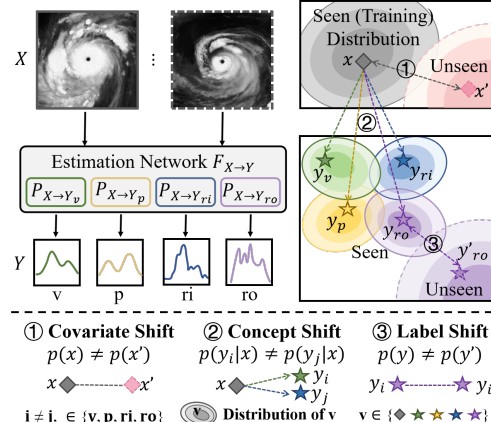

Figure 1: Diverse Distribution Shifts in TC Multi-Task Estimation. The v, p, ri and ro represent the wind speed, pressure, inner-core and outer-core size of TC, respectively.

**Contributions.** As illustrated in Figure 2, to cope with various distribution shifts, we propose a novel perspective based on Gaussian modeling that characterizes the invariant distributions of the feature embeddings, rather than the feature embeddings themselves. Although the environmental fields outside TCs are highly variable, the internal physical relationships among multiple TC attributes [15, 16] remain relatively stable and invariant. Consequently, for **C1**, we propose the **I**entity **D**istribution-**O**riented Physical Invariant **L**earning framework (**IDOL**), which imposes identity-oriented constraints on embeddings by regulating their distributions under the guidance of prior physical knowledge. The task-shared and task-specific identity tokens derived from the proposed IDOL represent the underlying physical invariant distribution in TC multi-task estimation.

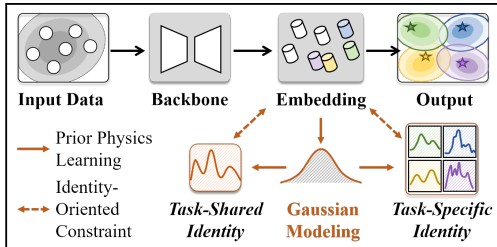

Figure 2: Solution to various distribution shifts: Regulating the feature space to task identities by prior physics.

For **C2**, the learned task-specific identity tokens transform the synchronous learning of conditional probabilities for multiple specific tasks $Y_i, i \in \{v, p, ri, ro\}$ into a task flow learning process, involving $P(Y_i|X)$ and $P(Y_j|Y_i)$, thereby addressing concept shift through the decoupling and modeling of task dependencies. The v, p, ri, and ro represent the estimated TC attributes, namely wind speed, pressure, inner-core size, and outer-core size, respectively. Meanwhile, the task-shared identity token is employed to address covariate and label shifts by linking the input and output and maximizing their information representation. Specifically, we incorporate two types of prior physics, including TC wind field model and dark correlation knowledge, to learn the physical invariance of TC, the major contributions of the proposed IDOL are summarized as follows:

- To address concept shift in multi-task learning, we propose a **Task Dependency Flow learning** module. By incorporating the prior wind field model, the conditional probabilities of multiple specific tasks are decoupled to model the dependencies among tasks, thereby facilitating the learning of distinct task-specific identities.
- To address covariate and label shifts, we design a **Correlation-Aware Information Bridge** module. By incorporating physical correlations to regulate the latent feature distribution, the task-shared identity token is modeled to serve as an information bridge that preserves the core information of both input and output in TC estimation.
- Comprehensive experiments are conducted on multiple TC estimation and prediction tasks to evaluate the effectiveness of the proposed IDOL. The results demonstrate the efficacy of

IDOL in handling diverse distribution shifts through feature space constraints informed by prior physical knowledge.

## 2 Related Work

**TC Estimation.** TC estimation is a thriving research field that aims to model and understand the intricate development dynamics in multiple attributes of TC. Since traditional methods [1, 5, 6] depend on the experiences of experts, numerous approaches for TC intensity estimation based on Convolutional Neural Network (CNN) [17–21] have been introduced. Similarly, various CNN-based methods have been employed to estimate multiple sizes of TC [22, 23, 4]. Another important direction involves leveraging multi-task learning to simultaneously capture the relationships between different TC attributes and estimate them [7, 8, 24–26]. These models employ the method of hard parameter sharing to capture the commonality of different tasks, facilitating TC multi-task estimation. However, most existing TC estimation models only rely on data-driven learning, limiting their generalization across diverse data distributions.

**Invariant Learning.** Invariant learning is a method aimed at learning features that are consistent for a specific task, while remaining robust to changes in environments or domains [27–29]. This approach tackles distribution shift issues, allowing models to generalize across varying conditions. One common method is to partition the training data into multiple environments or domains and optimize a shared objective function across these environments, ensuring that learned features are consistent across all of them [30, 31], thus mitigating distribution shifts. Another approach uses the Mixture of Experts framework, which decomposes the model into expert modules with dynamic routing mechanisms to separate domain-specific and domain-invariant features [32–34], improving cross-domain generalization.

**Physics-incorporated Methods.** There has been a recent surge of interest in combining physics with deep learning methods for improving the interpretability, and generalization of models. According to the previous survey [35], there are three different ways to inform deep learning models with physics. The first is observational bias [36, 37], which incorporates physics principles and constraints implicitly through data. The second one is learning bias approaches [38, 39], which incorporate the governing Partial Differential Equations more explicitly into the loss of training process. At last, in contrast to enforcing physical constraints, inductive bias approaches [40, 41] attempt to embed prior physics knowledge within the network design.

Since previous methods of invariant learning rely on environmental constraints and post-intervention output distribution, they can not cover diverse shifted distributions and are not suitable for specific meteorological tasks. In this work, to tackle diverse distribution shifts, we integrate observational bias and inductive bias to learn the identity distribution of tasks that represent physical invariance.

## 3 Preliminaries

**Representation of Correlated Data:** The specific semantic information of TC is captured and encoded using temporal-spatial satellite data and auxiliary physical data, collectively denoted as $\mathbf{X}$. For satellite data, we use multi-channel infrared brightness temperature data, which is denoted as $\mathbf{X}_{\mathtt{ir}} = \{\mathbf{X}_{\mathtt{ir}}^{\mathtt{t_0}}, \mathbf{X}_{\mathtt{ir}}^{\mathtt{t_1}}\}$, where $\mathbf{X}_{\mathtt{ir}} \in \mathbf{R}^{\mathtt{c} \times \mathtt{h} \times \mathtt{w}}$ at each time is a three-dimensional tensor. In this representation, $\mathtt{c}$ represents the number of channels, $\mathtt{h}$ and $\mathtt{w}$ denotes the height and width. Moreover, combined with the previous research in TC estimation [7, 42], we define the auxiliary data as $\mathbf{X}_{\mathtt{aux}} = \{\mathbf{X}_{\mathtt{dev}}, \mathbf{X}_{\mathtt{cor}}\}$, including developmental and correlation factors of TC. In this representation, $\mathbf{X}_{\mathtt{dev}}$ includes the previous level and the time since the TC became a named storm in minutes. $\mathbf{X}_{\mathtt{cor}}$ contains TC fullness, concentration ratio, energy ratio, and TC width. To ensure real-time applicability, all auxiliary data are collected from the 12 hours preceding the estimation time.

**Identity-Oriented TC Estimation.** In the context of TC multi-task estimation, a common goal is to estimate multiple current attributes by leveraging task-shared features extracted from multi-modal data. As shown in Eq. (1), the conventional objective is to learn a feature extractor $f$ and task estimators $g$ that minimize the overall estimation error:

$$\theta^* = \operatorname*{argmin}_{\theta} \mathcal{L}_{\mathtt{e}}\big(g(f(\mathbf{X}; \theta_1); \theta_2), \mathbf{Y}\big), \quad \theta = \{\theta_1, \theta_2\} \tag{1}$$

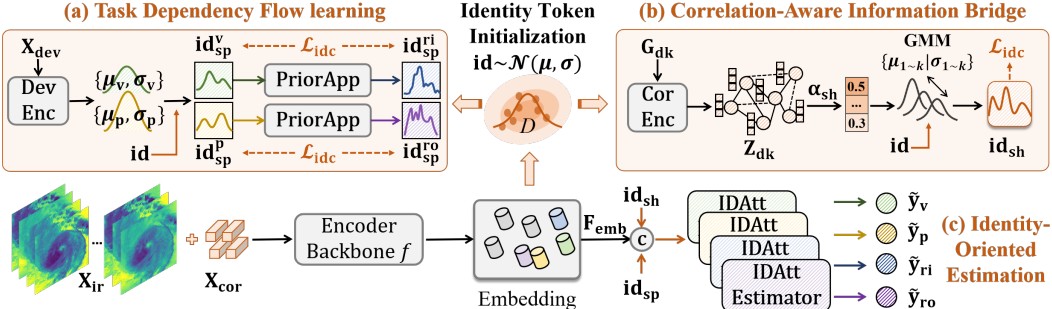

Figure 3: The framework of the proposed IDOL. `v`, `p`, `ri`, and `ro` denote the TC wind speed, pressure, inner-core size, and outer-core size, respectively. DevEnc, CorEnc, and GMM refer to the proposed Development Encoder, Correlation Encoder, and Gaussian Mixture Model, respectively.

where $\mathcal{L}_e$ denote the error loss function for all task, $\mathbf{X}$ and $\mathbf{Y}$ denote the input data and ground truth labels, respectively. However, this objective relies on the in-distribution assumption, which is often violated in practice, thereby limiting the model's generalization capability. To address diverse distribution shifts, we propose an identity-oriented TC estimation framework that captures physical invariance, as formally defined in Eq. (2).

$$\theta^* = \arg\min_\theta \underbrace{\mathcal{L}_e\big(g(\mathbf{id}_{\text{sp}}, \mathbf{id}_{\text{sh}}, f(\mathbf{X};\theta_1);\theta_2), \mathbf{Y}\big)}_{\text{Estimation Loss}} + \underbrace{\mathcal{L}_{\text{idc}}\big(\mathcal{O}_{\text{id}} \cdot D(f(\mathbf{X};\theta_1))\big)}_{\text{Identity-Oriented Constraint}} \tag{2}$$

where $\mathbf{id}_{\text{sp}}$ and $\mathbf{id}_{\text{sh}}$ refer to the proposed task-specific and task-shared identity tokens. $\mathcal{L}_{\text{idc}}$ is the identity-oriented constraint imposed on the feature distribution $D(f(\mathbf{X};\theta_1))$. Specifically, we design the operator $\mathcal{O}_{\text{id}}$ to model task identity tokens by leveraging the internal invariant physical correlations to regulate the feature distribution. In our method, the function $g$ refers to the proposed estimation heads based on the Correlation-Aware Attention mechanism. And $\theta_1$ refers to the parameters of the encoder backbone, while $\theta_2$ includes the parameters of the modules illustrated in Figure 3(a)-(c).

# 4 Our Approach: IDOL

This section provides an in-depth explanation of the technical details of the proposed IDOL framework. The goal of our method is to achieve identity-oriented estimation of multiple TC attributes, including wind speed, pressure, and inner- and outer-core sizes, denoted by the subscripts `v`, `p`, `ri`, and `ro`, respectively. As illustrated in Figure 3, we present a comprehensive overview of the framework. Based on the feature embedding $\mathbf{F}_{\text{emb}}$ extracted by the encoder backbone $f$, the identity token $\mathbf{id}$, which represents the physical invariant distribution of TC, is initialized as follows:

$$\mathbf{F}_{\text{emb}} = f(\mathbf{X};\theta_1) \; ; \; \mathbf{id} = \mathcal{P}_{\text{gaussian}}(\mathbf{F}_{\text{emb}}) \tag{3}$$

$\mathcal{P}_{\text{gaussian}}$ denotes a Gaussian sampling operation applied based on the mean and variance of the feature embeddings. Details of the Eq. (3) are provided in Appendix B.1.

Then, based on the initialized identity, the proposed model consists of three main components for handling distribution shifts: (a) **Task Dependency Flow Learning:** To address concept shift in multi-task learning, we introduce a Development Encoder and a Prior-aware Approximator (PriorApp) to decouple and model task dependencies. This facilitates the learning of all task-specific identity tokens, denoted as $\mathbf{id}_{\text{sp}} = \{\mathbf{id}_{\text{sp}}^{\text{v}}, \mathbf{id}_{\text{sp}}^{\text{p}}, \mathbf{id}_{\text{sp}}^{\text{ri}}, \mathbf{id}_{\text{sp}}^{\text{ro}}\}$. (b) **Correlation-Aware Information Bridge:** To handle covariate and label shifts in the input and output, we construct a dark knowledge graph to learn latent physical correlations among TC attributes. These learned correlations are used to regulate the embedding distribution. The resulting task-shared identity, denoted as $\mathbf{id}_{\text{sh}}$, serves as a bridge between input and output by capturing their invariant physical relationships. (c) **Identity-Oriented Estimation:** The feature embedding, along with the task-specific and task-shared identity tokens, is fed into a multi-head self-attention mechanism to impose identity-oriented constraints. Finally, four estimation heads produce the outputs based on the identity-oriented feature embedding.

### 4.1 Task Dependency Flow Learning

In this section, to decouple and model task dependencies, we first assume that the development of TC is in line with the prior Holland model shown in Eq. (4). In this prior model, an inherent physical dependency exists between the intensity and wind radii of TC, in which, intensity includes wind speed and pressure, and wind radii include the size of the inner and outer core. Therefore, to model the task-specific identity tokens, we decompose the task dependency flow learning into two sequential modules: Development Encoder and Prior-aware Approximator, which are responsible for modeling the identity of TC intensity and size, respectively.

**Prior Wind Field Model.** To model task-specific identity tokens by the decoupling and modeling of task dependencies, we incorporate the Holland model into the framework, which is a classical TC wind field model, providing a mathematical framework for describing the radial wind distribution of TC [15]. Based on quasi-statistical physical assumptions, this model assumes a quasi-steady-state balance, where the wind field is driven by the interplay of dynamic and thermodynamic processes, reflecting a balance between pressure gradient force, centrifugal force, and Coriolis force. The formulation can be represented as follows:

$$r^B \ln[(p_\mathrm{n} - p_\mathrm{c})/(p_\mathrm{r} - p_\mathrm{c})] = A \tag{4}$$

where $p_\mathrm{r}$ is the pressure at radius $r$, $p_\mathrm{c}$ is the central pressure and $p_\mathrm{n}$ is the ambient pressure (theoretically at infinite radius). The $A$ and $B$ are scaling parameters, which allow the model to adapt to different types of TC by adjusting the steepness of the wind profile.

**Development Encoder.** Considering TC wind speed and pressure are highly related to the state of TC, which reflects the dynamic development. Consequently, we design an encoder to explore the developmental mode from the developmental factors of TC, and use the mode features to regulate the initialized identity token ($\mathbf{id}$), then obtain the task-specific identity tokens of TC wind speed and pressure (denoted as $\mathbf{id}_\mathrm{sp}^\mathrm{v}$ and $\mathbf{id}_\mathrm{sp}^\mathrm{p}$). The above processes can be expressed as follows:

$$\boldsymbol{\mu}_\mathtt{i}, \boldsymbol{\sigma}_\mathtt{i} = \mathbf{Enc}_\mathrm{dev}(\mathbf{X}_\mathrm{dev}; \theta_\mathrm{dev}) \; ; \; \mathbf{id}_\mathrm{sp}^\mathtt{i} = \boldsymbol{\mu}_\mathtt{i} + \boldsymbol{\sigma}_\mathtt{i} \cdot \mathbf{id}, \mathtt{i} \in \{\mathtt{v}, \mathtt{p}\} \tag{5}$$

where $\mathbf{Enc}_\mathrm{dev}$ denotes the proposed Development Encoder, composed of linear layers followed by ReLU activations, with corresponding parameters $\theta_\mathrm{dev}$. The operator $\mathcal{O}_\mathrm{id}$ defined in Eq. (2) is implemented here via the reparameterization formulation: $\boldsymbol{\mu}_\mathtt{i} + \boldsymbol{\sigma}_\mathtt{i} \cdot \mathbf{id}$.

**Prior-aware Approximator.** Through an identical transformation of Eq. (4), we derive the transcendental equation in Eq. (6), with its derivation provided in Appendix B.2:

$$\mathbf{r} = \left(\frac{A}{\ln(p_\mathrm{n} - p_\mathrm{c}) - \ln(p_\mathrm{r} - p_\mathrm{c})}\right)^{\frac{1}{B}} = \boldsymbol{\gamma}^{\frac{1}{B}} \tag{6}$$

To encode the prior task dependency captured by Eq. (6), we propose the PriorApp, which transforms task-specific identity tokens across tasks via two key components: (i) learnable linear mappings to approximate the intermediate term $\boldsymbol{\gamma}$, and (ii) a deep iterative algorithm that simulates the nonlinear exponentiation, with each iteration defined in Eq. (16) of Appendix B.2. As an illustrative example, the dependency between TC pressure and outer-core size is modeled as follows:

$$\begin{aligned}
\gamma_\mathrm{ro} &= \frac{\alpha_\mathrm{p}^\mathrm{ro}}{\ln\left(\mathcal{F}_\mathrm{n}(\mathbf{id}_\mathrm{sp}^\mathrm{p}; \theta_\mathrm{n})\right) - \ln\left(\mathcal{F}_\mathrm{r}(\mathbf{id}_\mathrm{sp}^\mathrm{p}; \theta_\mathrm{r})\right)} \\
\mathbf{U} &= \mathrm{GConv}(([\mathbf{U}_1, \mathbf{U}_2, \ldots, \mathbf{U}_n], \mathbf{E}_\mathrm{u}); \theta_\mathrm{u}) \\
\mathbf{id}_\mathrm{sp}^\mathrm{ro} &= \mathbf{H}^\mathtt{t} := \mathrm{PriorApp}^\mathtt{t}(\mathbf{H}^{\mathtt{t}-1}, \mathbf{U}, \boldsymbol{\gamma}_\mathrm{ro}; \theta_\mathrm{iter}), \; \text{s.t. } \delta(\mathbf{H}^\mathtt{t}, \mathbf{H}^{\mathtt{t}-1}) < \tau
\end{aligned} \tag{7}$$

where $\ln(\cdot)$ denotes the natural logarithm; $\mathcal{F}_\mathrm{n}$ and $\mathcal{F}_\mathrm{r}$ are linear layers with learnable parameters $\theta_\mathrm{n}$ and $\theta_\mathrm{r}$; and $\alpha_\mathrm{p}^\mathrm{ro}$ is a learnable scaling factor. $\mathbf{U}_\mathrm{i}$ represents the learnable node embeddings, and $\mathbf{E}_\mathrm{u}$ is their edge connectivity matrix, where an edge exists if two nodes are adjacent. The GConv here refer to the Graph Convolutional Layer, which is introduced to supplement the prior equation by modeling the additional latent interactions through $\mathbf{U}$. The deep iterative algorithm terminates either upon reaching a predefined maximum number of iterations or when the difference $\delta(\mathbf{H}^\mathtt{t}, \mathbf{H}^{\mathtt{t}-1})$ falls below a threshold $\tau$.

## 4.2 Correlation-Aware Information Bridge

To address covariate and label shifts in the input and output, we propose modeling the task-shared identity token $\mathbf{id}_{\mathtt{sh}}$, which acts as an information bridge between them by capturing invariant physical relationships. Inspired by the concept of dark knowledge proposed in LoCa [44], which encompasses multi-task information, we first construct a dark knowledge graph $\mathbf{G}_{\mathtt{dk}}$ to uncover latent physical correlations among TC attributes and input data. Details of the construction of $\mathbf{G}_{\mathtt{dk}}$ are provided in Appendix B.3. Then, to model the task-shared identity token grounded in these correlations, we use the learned graph to inform the regulation of feature distribution. Specifically, the proposed module comprises two components: (i) a correlation encoder (CorEnc) using stacked GConv layers to capture latent correlations, and (ii) a self-adaptive Gaussian Mixture Model (GMM) with correlation-aware weighting. The GMM flexibly combines latent distributions to effectively capture shared identity features across tasks. The learning process is formulated as follows:

$$\mathbf{Z}_{\mathtt{dk}} = \mathbf{Enc}_{\mathtt{cor}}(\mathbf{G}_{\mathtt{dk}}; \theta_{\mathtt{dk}}) \; ; \boldsymbol{\alpha}_{\mathtt{sh}} = \mathbf{Softmax}(\mathbf{Z}_{\mathtt{dk}})$$
$$\mathbf{id}_{\mathtt{sh}} = \boldsymbol{\alpha}_{\mathtt{sh}} \cdot \mathbf{Aggregate}(\{\mathbf{d}_0, ..., \mathbf{d}_k | \mathtt{j} = 0, 1, ..., k\}) \, , \mathbf{d}_{\mathtt{j}} = \boldsymbol{\mu}_{\mathtt{j}} + \boldsymbol{\sigma}_{\mathtt{j}} \cdot \mathbf{id} \tag{8}$$

Here, $\mathbf{Enc}_{\mathtt{cor}}$ denotes the Correlation Encoder with parameters $\theta_{\mathtt{dk}}$. The GMM contains $k$ learnable components $\{\mathbf{d}_{\mathtt{j}}\}_{\mathtt{j}=0}^{k}$, each constructed from a parameter pair $(\boldsymbol{\mu}_{\mathtt{j}}, \boldsymbol{\sigma}_{\mathtt{j}})$.

## 4.3 Identity-Oriented Estimation

To enable identity-oriented TC estimation, we design four estimation heads based on correlation-aware attention [45] to output all TC attributes. These heads are collectively referred to as the **IDAtt Estimator** in Figure 3. The estimation process and final loss based on Mean Absolute Error (MAE, denoted as $\mathcal{L}_{\mathtt{MAE}}$) and identity-oriented constraint $\mathcal{L}_{\mathtt{idc}}$ are given by Eq. (9):

$$\widetilde{\mathbf{y}}_{\mathtt{i}} = \Phi_{\mathtt{id}}^{\mathtt{i}}(\mathcal{P}_{\mathtt{cat}}(\mathbf{id}_{\mathtt{sp}}^{\mathtt{i}}, \mathbf{id}_{\mathtt{sh}}, \mathbf{F}_{\mathtt{emb}}); \theta_{\phi}^{\mathtt{i}}), \mathtt{i} \in \{\mathtt{v}, \mathtt{p}, \mathtt{ri}, \mathtt{ro}\}$$
$$\mathcal{L}_{\mathtt{e}} = \sum \mathcal{L}_{\mathtt{MAE}}(\widetilde{\mathbf{y}}_{\mathtt{i}}, \mathbf{y}_{\mathtt{i}}) \, ; \; \mathcal{L}_{\mathtt{idc}}(\mathbf{id}, \mathbf{y}) = \mathcal{L}_{\mathtt{MAE}}(\mathbf{id} \cdot \mathbf{id}^{\mathtt{T}}, \mathbf{y} \cdot \mathbf{y}^{\mathtt{T}}) \tag{9}$$
$$\mathcal{L}_{\mathtt{total}} = \mathcal{L}_{\mathtt{e}} + \lambda \cdot (\mathcal{L}_{\mathtt{idc}}(\mathbf{id}_{\mathtt{sh}}, \mathbf{Y}) + \sum \mathcal{L}_{\mathtt{idc}}(\mathbf{id}_{\mathtt{sp}}^{\mathtt{i}}, \mathbf{y}_{\mathtt{i}}))$$

where $\Phi_{\mathtt{id}}^{\mathtt{i}}$ consists of a transformer encoder with correlation-aware attention, followed by three linear layers with ReLU activations. $\widetilde{\mathbf{y}}_{\mathtt{i}}$, $\mathbf{y}_{\mathtt{i}}$ and $\mathbf{Y}$ denote the estimated value, the ground-truth value of the specific task, and the concatenation of ground-truth values across all TC attributes, respectively. The $\mathcal{L}_{\mathtt{idc}}$ encourages alignment between the intra-sample correlations of the learned identity and those of the corresponding ground-truth labels.

## 5 Experiments

In this section, we first illustrate the datasets and experiments settings of this paper. Then, comprehensive experiments, including comparison experiments, ablation experiments, and the analytical experiments were conducted to answer the following questions. **Q1**: How does the proposed IDOL framework perform in TC estimation and even forecasting tasks? **Q2**: Do the learned identity tokens effectively capture the physical invariance of TCs? Specifically, does identity-oriented estimation enhance both the accuracy and the distribution alignment of TC attribute estimations by addressing all types of distribution shifts? **Q3**: How does the proposed IDOL framework address concept, covariate, and label shifts using task-shared and task-specific identity tokens?

### 5.1 Experimental Setup

**Datasets.** To evaluate how well the proposed model performs in TC estimation, our newly constructed Physical Dynamic TC datasets (PDTC) and the Digital TC datasets [46] are used. These datasets include records of 303 TCs from 2015 to 2023 over the Western North Pacific, containing various TC-related information, such as position, age, and corresponding satellite cloud images. Notably, the PDTC dataset incorporates additional TC correlation factors, e.g. TC fullness, to better capture the dynamic development of TC. Details of the datasets are provided in Appendix C.1.

**Metrics.** To assess the performance of the model in both TC estimation and prediction, including trajectory (distance, km), wind radii (nmi), pressure (hPa), and wind speed (m/s), we employ two

Table 1: Comparison of TC multi-task estimation methods on the Physical Dynamic TC datasets.

| Categories | Comparison Methods | Wind Speed | | | Pressure | | | Inner-Core Size | | | Outer-Core Size | | |
|---|---|---|---|---|---|---|---|---|---|---|---|---|---|
| | | MAE | RMSE | STD | MAE | RMSE | STD | MAE | RMSE | STD | MAE | RMSE | STD |
| Traditional | ADT | 11.2 | 14.2 | - | 8 | 10.2 | - | - | - | - | - | - | - |
| | MTCSWA | - | - | - | - | - | - | 11.7 | 18.2 | - | 26.4 | 33 | - |
| Multi-Modal Fusion | STIA | 10.7 | 14.41 | 9.67 | - | - | - | - | - | - | - | - | - |
| | NS | - | - | - | - | - | - | 10.5 | 15.51 | 11.44 | 26.35 | 36.12 | 24.71 |
| | TC-MTLNet | 13.82 | 18.06 | 11.62 | 12 | 15.46 | 9.79 | - | - | - | 31.49 | 42.83 | 29.14 |
| | DeepTCNet | 8.84 | 11.76 | 7.76 | 8.13 | 10.42 | 5.31 | 8.09 | 13.71 | 11.25 | 25.86 | 33.49 | 21.29 |
| | PeRCNN | 10.04 | 13.23 | 8.61 | 6.99 | 8.78 | 6.52 | 8.56 | 14.13 | 11.25 | 24.97 | 32.94 | 21.48 |
| Invariant Learning | IRM | 9.54 | 12.93 | 8.71 | 7.71 | 10.06 | 6.47 | 8.12 | 13.7 | 11.06 | 25.6 | 34.7 | 23.37 |
| | V-Rex | 10 | 13.29 | 8.76 | 8.13 | 10.48 | 6.6 | 8.21 | 13.7 | 10.94 | 25.4 | 34 | 22.59 |
| | SADE | 10.01 | 13.47 | 8.66 | 8.09 | 10.55 | 6.76 | 8.23 | 13.41 | 10.59 | 25.95 | 34.19 | 22.26 |
| | DirMixE | 9.93 | 13.27 | 8.8 | 7.92 | 10.27 | 6.54 | 8.47 | 14.2 | 11.37 | 25.5 | 34.3 | 23.02 |
| | **IDOL** | **5.93** | **7.6** | **4.75** | **5.77** | **7.15** | **4.23** | **6.24** | **12.06** | **10.31** | **17.06** | **23.26** | **15.8** |

widely-used evaluation metrics: Mean Absolute Error (MAE) and Root Mean Squared Error (RMSE). The lower values of these metrics indicate better performance. Moreover, the standard deviation (STD) of error on test sets is selected as an additional metric to measure the stability and generalization ability of the model.

**Baselines.** To evaluate the effectiveness of the proposed, we compare our IDOL with several representative SOTA methods, which can be divided into three groups:

- **Traditional Methods.** ADT and Multiplatform Tropical Cyclone Surface Wind Analysis technique (MTCSWA) are typical methods for using satellite infrared data to estimate TC intensity and wind radii respectively [53, 6].
- **Multi-Modal Fusion Methods.** We select several typical TC estimation methods based on multi-modal data understanding, including single-task [21, 23] and multi-task estimation [8, 7, 54]. Additionally, PeRCNN is a physical method that models polynomial correlations, which can be adapted for TC estimation. Specifically, following the correlation modeling method in Phy-CoCo [55], we use PeRCNN to transform the task-specific identities of TC attributes, which are then compared in subsequent analysis experiments of task-specific identity distributions.
- **Invariant Learning Methods.** Invariant learning methods, including standard IRM [30], V-REx [31] and MoE based methods SADE and DirMixE [32, 33].

## 5.2 Overall Performance (for Q1)

Table 1 and Table 5 reports the overall performance of all the methods on PDTC datasets and Digital TC datasets respectively, where the best is shown in bold. Among the SOTA methods, DeepTCNet, based on observational bias, and PeRCNN, based on inductive bias, generally perform well. This suggests that leveraging the physical relationships among multiple TC attributes can effectively improve TC estimation. However, general invariant learning methods perform poorly, primarily due to the lack of domain-specific knowledge integration. In contrast, the proposed IDOL incorporates prior knowledge into the feature distribution to model identity tokens with physical invariance, achieving the best performance among all SOTA methods. Additionally, Figures 9 and 10 in Appendix C.2 further illustrate the robust estimation results across TC categories on unseen TCs in both the PDTC and Digital TC datasets. Specifically, for estimating wind speed, pressure, inner-core size, and outer-core size on the PDTC dataset, IDOL outperforms the previous best method, DeepTCNet, by 32.9%, 29%, 22.9%, and 34% in MAE, respectively.

The above experimental results highlight the importance of modeling task identity tokens with physical invariance, as it enables the model to generalize effectively to previously unseen TC. As shown in Table 6 and 7 in Appendix C.2, our model achieves the best estimation performance without increasing model parameters or inference time. Moreover, as shown in Table 9 in the Appendix C.2, the improvement in TC forecasting accuracy with the proposed method further proves the ability of our IDOL to handle distribution shifts, which are also inevitable in TC forecasting.

## 5.3 Identity-Oriented Performance (for Q2)

**Ablation Study.** In this section, as shown in Table 2, we conduct an ablation study to investigate the contributions of physical identity tokens in the proposed IDOL. First, we observe that the model with

Table 2: Ablation experiments. $\mathbf{Net}_f$ refers to the encoder backbone $f$. $\mathbf{id}_{sp}$ and $\mathbf{id}_{sh}$ represent task-specific and task-shared identity tokens with physical invariance, addressing concept shift and covariate/label shifts, respectively.

| $\mathbf{Net}_f$ | $\mathbf{id}_{sp}$ | $\mathbf{id}_{sh}$ | Wind Speed | | | Pressure | | | Inner-Core Size | | | Outer-Core Size | | |
|---|---|---|---|---|---|---|---|---|---|---|---|---|---|---|
| | | | MAE | RMSE | STD | MAE | RMSE | STD | MAE | RMSE | STD | MAE | RMSE | STD |
| ✓ | | | 10.13 | 13.25 | 8.54 | 7.79 | 10.13 | 6.47 | 8.32 | 13.87 | 11.1 | 28.58 | 37.66 | 24.52 |
| ✓ | ✓ | | 7.24 | 9.13 | 5.55 | 6.66 | 8.27 | 4.97 | 7.37 | 13.24 | 10.99 | 24.91 | 33.28 | 22.07 |
| ✓ | ✓ | ✓ | **5.93** | **7.6** | **4.75** | **5.77** | **7.15** | **4.23** | **6.24** | **12.06** | **10.31** | **17.06** | **23.26** | **15.8** |

the proposed identity tokens derived from prior physical knowledge, significantly improve both the accuracy and stability of the model across the four estimation tasks. Secondly, as shown in Table 8 in Appendix C.3, the model with task-specific identity tokens learned by the prior Holland model outperforms those learned by a noisy prior or linear layers. This further validates the importance of incorporating effective prior knowledge. Since the test and training datasets are completely non-overlapping in the temporal domain, meaning the test set comprises entirely unseen TCs, which effectively represents an unknown data distribution. Therefore, in addition to evaluating estimation accuracy, we also visualize the distributions of ground-truth and estimated results on the test set using kernel density estimation (KDE). As shown in Figure 4(a), the distributions of TC attributes estimated by our IDOL are closer to the true distributions compared to baseline models such as DeepTCNet and PeRCNN. This demonstrates that the learned physical identity tokens enable the accurate estimations in terms of both error and distribution alignment.

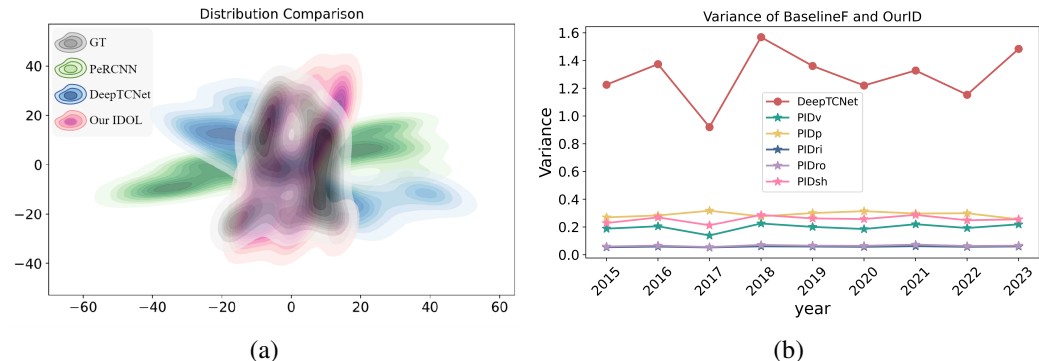

(a)  (b)

Figure 4: (a) Distribution visualization of test set estimation results based on KDE. (b) Variance comparison between physical identity (PID) tokens and the features extracted by DeepTCNet.

Moreover, to explore how different degrees of parameter sharing affect model size and multi-task performance, we conducted ablation experiments by varying the parameter ratios between task-shared and task-specific identity tokens. Specifically, we define the ratio based on the dimensional size of the corresponding identity token, with configurations including 1:1, 1:2, 1:3, 2:1, and 3:1. For example, a ratio of 1:2 means that the task-shared identity token has half the dimensionality of the task-specific identity token. The results are presented in Table 3.

Table 3: Additional ablation experiments on the ratios between task-shared and task-specific identity tokens. Bold indicates the best result, and underline denotes the second best.

| Ratio Settings ($\mathbf{id}_{sh} : \mathbf{id}_{sp}$) | Wind Speed | | | Pressure | | | Inner-Core Size | | | Outer-Core Size | | |
|---|---|---|---|---|---|---|---|---|---|---|---|---|
| | MAE | RMSE | STD | MAE | RMSE | STD | MAE | RMSE | STD | MAE | RMSE | STD |
| 1:2 | **5.75** | **7.46** | 4.76 | 5.78 | 7.17 | 4.25 | 6.36 | 12.24 | 10.45 | 17.95 | 24.74 | 17.02 |
| 1:3 | 6.1 | 7.85 | 4.95 | 5.95 | 7.36 | 4.33 | 6.3 | 12.07 | 10.29 | 18.11 | 24.52 | 16.53 |
| 2:1 | 5.91 | 7.58 | **4.75** | 5.9 | 7.33 | 4.35 | 6.32 | 12.03 | 10.24 | 17.94 | 24.52 | 16.72 |
| 3:1 | 5.93 | 7.69 | 4.89 | 5.79 | 7.14 | **4.17** | **6.13** | **11.81** | **10.09** | 18.79 | 25.34 | 17 |
| 1:1 | 5.93 | 7.6 | **4.75** | **5.77** | 7.15 | 4.23 | 6.24 | 12.06 | 10.31 | **17.06** | **23.26** | **15.8** |

As shown in Table 3, under the given model design and experimental settings, varying the parameter ratio between task-shared and task-specific identity tokens has slight impact on model size and overall estimation performance. This is mainly because the proposed identity tokens, even with lightweight configurations, are sufficient to impose physical constraints on the feature distributions, thereby guiding the model to effectively capture the intrinsic characteristics required for multi-task

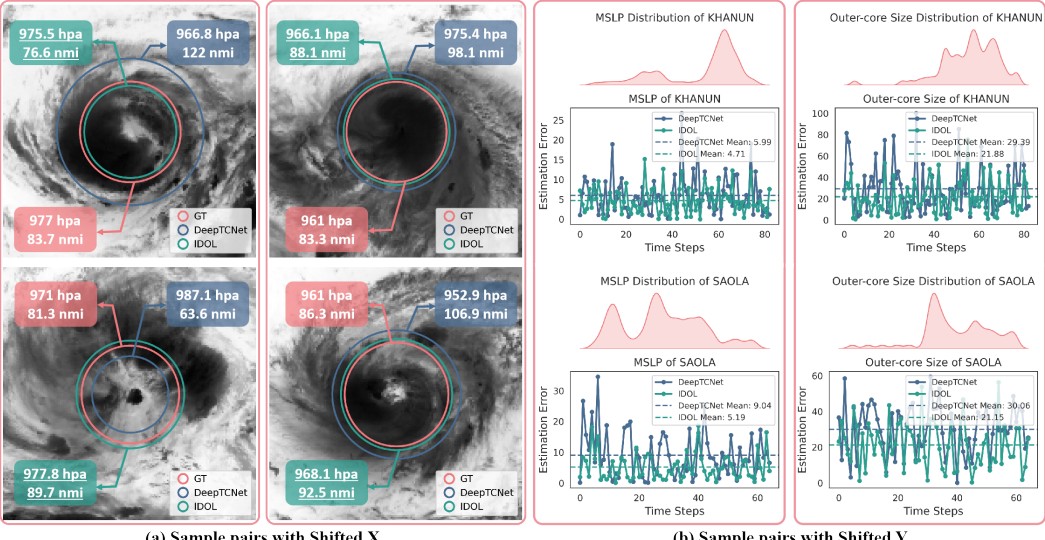

(a) Sample pairs with Shifted X                    (b) Sample pairs with Shifted Y

Figure 5: Estimation performance of pressure and outer-core size on test samples under distribution shifts: (a) Estimation results under covariate shift and (b) Mean absolute errors under label and concept shifts across different TCs and output attributes. Each vertical red box indicates a pair of samples exhibiting distribution shift.

learning. Increasing the dimensionality of a specific identity token primarily enables more fine-grained representation learning, which may lead to slight variations in task-wise performance, but does not result in significant overall performance gains. This further confirms the effectiveness and efficiency of the proposed identity token design in ensuring consistent and robust multi-task learning.

**Invariant Learning.** To answer **Q2**, we visualize the statistical variance of the proposed physical identity tokens and DeepTCNet features in Figure 4(b) to assess whether the learned identities capture TC invariance. In invariant learning, features from different domains should be close to each other, which means they will have similar variances. Supposed the model extracts physically invariant identity tokens across time domains, their statistical variance should remain consistent. From Fig 4(b), compared to DeepTCNet features, our physical identity tokens exhibit smaller variance across domains, indicating stronger robustness to domain shifts. This improvement supports our assumption: by leveraging physical correlations to constrain feature distributions, the learned identities capture the intrinsic physical invariance of TCs. In other words, incorporating prior physics enables the model to better understand the internal TC dynamics and generalize to unseen, variable environments.

Moreover, to evaluate the model's robustness under distribution shifts, we screen out and visualize specific sample pairs from the test set that exhibit distribution shifts in both input and output. Following the task dependencies in the prior Holland model, pressure and outer-core size are treated as a correlated task group, their shifted sample results shown in Figure 5. Similarly, the results for wind speed and inner-core size are presented in Figure 13 in Appendix C.4. The details of the selected shifted sample pairs are also provided in Appendix C.4. In Figure 5, three types of distribution shifts are illustrated: covariate shift, referring to difference in TC structures as seen in the input; label shift, referring to distribution differences of the same output attribute across different TCs; and concept shift, referring to distribution differences across different output TC attributes. Compared with SOTA method DeepTCNet, in Figure 5(a), the estimations of our IDOL (green boxes with underlines) are noticeably closer to the ground truth (red boxes), even under covariate shift. Meanwhile, in Figure 5(b), IDOL also achieves lower estimation error across different TC instances and attributes with varying distributions, showing its effectiveness under label and concept shifts. These results collectively demonstrate IDOL's robustness to various distribution shifts, validating its capability in physical invariance learning.

## 5.4 Analysis of Physical Identity (for Q3)

To address **Q3**, we conduct analytical experiments on the learned task-specific and task-shared identity tokens. First, assuming that concept shift in multi-task estimation can be mitigated by decoupling

and modeling inter-task dependencies (i.e., the `v-ri` and `p-ro` relationships in the Holland model), we compute the mutual information (MI) between each true TC attribute and its corresponding task-specific identity token to quantify the adequacy of dependency modeling. As shown in Figure 6(a)-(b), the MI between task-specific identity tokens is even higher than that of between the ground-truth attributes themselves, on both seen (training) and unseen datasets. This indicates that the learned physical identities effectively model prior task dependencies, i.e., $P(Y_j|Y_i)$, as higher MI indicates stronger inter-variable correlations. Moreover, since task dependency flows are grounded in learning $P(Y_i \mid X)$, where $i \in \{v, p\}$, we further compute the MI between the output attribute and its corresponding identity token for tasks `v` and `p` to quantify the adequacy of correlation modeling. Figure 12 (Appendix C.5) presents this result, where the x-axis represents the degree of concept shift relative to the training set, and the y-axis shows the corresponding MI. The figure reveals that MI remains consistently high across varying shift degrees, with the average MI even higher on the test set than on the training set. This confirms that task-specific identities successfully preserve essential information for estimating `v` and `p`.

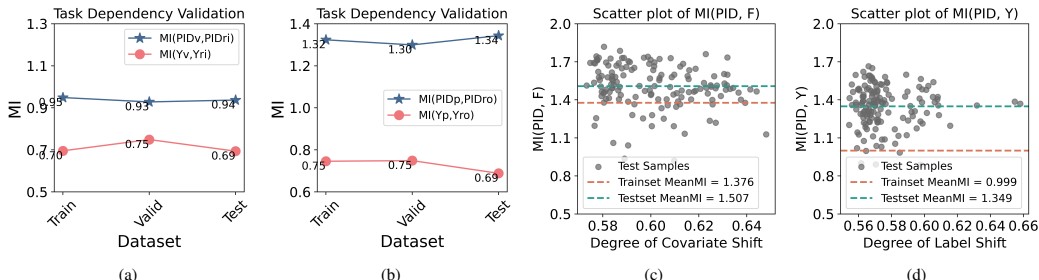

Figure 6: (a) Adequacy of modeling the task dependency between `v` and `ri`. (b) Adequacy of modeling the task dependency between `p` and `ro`. (c) Adequacy of correlation modeling between the learned task-shared identity token and input features F on the test set. (d) Adequacy of correlation modeling between the learned task-shared identity token and output attributes Y on the test set. PID and Y denote the proposed physical identity tokens and ground-truth TC attributes, respectively.

Second, to evaluate whether covariate and label shifts are mitigated by using the task-shared identity as a bridge between inputs and outputs, we compute the MI between the task-shared identity and both the input features and the output attributes, respectively. As shown in Figure 6(c)-(d), the MI remains high across varying degrees of shift, and the average MI on the test set is higher than that on the training set. This demonstrates that the task-shared identity effectively acts as a robust bridge, capturing consistent information under covariate and label shifts. Further evidence is provided in Figure 11 (Appendix C.5), where the estimation performance using real and randomized dark knowledge graphs is compared. The negligible difference in MAE and STD indicates that the proposed correlation-aware mechanism successfully captures meaningful physical invariances of TC.

# 6 Conclusions

This paper addresses the challenge of diverse distribution shifts to improve model robustness in TC estimation. The core idea is to learn invariant identity tokens that represent different task distributions by leveraging prior physics to constrain feature embeddings. Various distribution shifts are tackled by modeling prior task dependencies and latent physical correlations. Comprehensive experiments show that IDOL outperforms previous state-of-the-art methods, underscoring the effectiveness of prior physics modeling in handling unknown data distributions for TC estimation.

## Acknowledgments and Disclosure of Funding

This work is partially supported by Zhejiang Provincial Natural Science Foundation of China under Grant No. LRG25F020002, Distinguished Young Scholar of Shandong Province under Grant ZR2023JQ025 and Natural Science Foundation of China under Grant No. U24A20221 and No. 62202429.

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

# A  Distribution Shifts in TC.

To verify distribution shifts across different TC attributes, we use violin plots to visualize the distributions of input $X$ and output $Y$ across datasets or tasks. Using a standard normal distribution as reference, we compute the Jensen–Shannon divergence (JSD) between the input, output, and contrast distributions across datasets. The JSD values for each data batch are shown in the violin plots in Figure 7(a) and (b). The varying shapes of the violins in Figures 7(a) and (b) indicate clear differences in the distributions of $X$ and $Y$, confirming the presence of covariate and label shifts. Furthermore, Figure 7(c) presents the variance of ground-truth values across batches for each task, where the differences among tasks illustrate the existence of concept shift. These findings underscore the necessity of introducing physical invariant learning to model task-specific identity tokens, enabling the model to handle distribution shifts effectively and enhancing generalization.

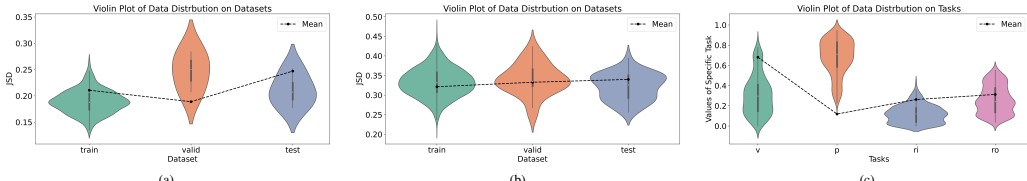

(a)                                             (b)                                             (c)

Figure 7: (a)-(b): Violin plots of JSD between input/output and contrast distributions across datasets, indicating covariate and label shifts. (c): Violin plots of real variance across batches for each task, indicating concept shift.

To explain the aforementioned distribution shifts, we summarize their definitions and underlying causes as follows:

- **Label shift.** The distribution of labels $P(Y)$ changes between the training and testing datasets, even if the conditional distribution $P(X \mid Y)$ remains unchanged. i.e. Due to climate change, including global warming and rising sea levels, the intensity and frequency of TCs may change over time, leading to label shifts.

- **Covariate shift.** The distribution of input $P(X)$ changes, while the conditional distribution $P(Y \mid X)$ remains unchanged. i.e. TC shapes may vary significantly due to geographical differences, reflected in brightness gradients and structural complexity, indicating a shift in the input distribution. However, the conditional distribution $P(Y \mid X)$, mapping cloud images to TC attributes, remains consistent.

- **Concept shift.** The conditional probability $P(Y \mid X)$ varies across different tasks. That is, influenced by distinct factors such as sea surface temperature or vertical wind shear, different TC attributes may follow distinct distributions and correlations. Thus, rules learned for one attribute (e.g., intensity) may not generalize to others, leading to concept shift.

# B  More details of IDOL

## B.1  Encoder Backbone

This section details the encoder backbone architecture employed in our framework. To enable identity-oriented estimation, it is essential to thoroughly analyze sequential satellite data and provide well-informed initializations for the identity tokens. To achieve this, our backbone is designed as a spatio-temporal semantic fusion network, comprising three main components: (i) Infrared Encoder: This module effectively captures features that are sensitive to both channel-wise and spatial variations by processing multi-channel infrared satellite data. (ii) Spatio-Temporal Semantic Fusion: This component introduces correlation-aware attention to guide the integration of features across time steps. It captures the interdependencies between temporal sequences and physical auxiliary inputs, enhancing the semantic richness of the learned representations. (iii) Gaussian Distribution Sampling: To initialize the identity tokens, we employ Gaussian distribution sampling based on the fused representations.

**Infrared Encoders.** VGG13 efficiently extracts multi-scale features through its convolutional layers, with initial layers capturing low-level features and deeper layers capturing complex semantics. Given

its suitability for capturing both local and global image features, we use VGG13 [47] as the encoder ($\mathbf{Enc}_{\mathrm{IR}}$) for satellite infrared data. The process of the proposed infrared encoders for satellite data is shown as follows:

$$\mathbf{F}_{\mathrm{ts}}^{\mathrm{t_0}} = \mathbf{Enc}_{\mathrm{IR}}(\mathbf{X}_{\mathrm{ts}}^{\mathrm{t_0}}; \theta_{\mathrm{IR}}^{\mathrm{t_0}}); \mathbf{F}_{\mathrm{ts}}^{\mathrm{t_1}} = \mathbf{Enc}_{\mathrm{IR}}(\mathbf{X}_{\mathrm{ts}}^{\mathrm{t_1}}; \theta_{\mathrm{IR}}^{\mathrm{t_1}})$$
$$\mathbf{F}_{\mathrm{ts}} = \mathbf{OP}_{\mathrm{cat}}(\mathbf{F}_{\mathrm{ts}}^{\mathrm{t_0}}, \mathbf{F}_{\mathrm{ts}}^{\mathrm{t_1}}) \tag{10}$$

where $\theta_{\mathrm{IR}}^{\mathrm{t_0}}$ and $\theta_{\mathrm{IR}}^{\mathrm{t_1}}$ are the parameters of encoders. $\mathbf{F}_{\mathrm{ts}} \in \mathbf{R}^{\mathrm{t} \times \mathrm{c} \times \mathrm{h} \times \mathrm{w}}$, where $t$ represents the number of time series.

**Spatio-Temporal Semantic Fusion.** Based on the multi-scale features $\mathbf{F}_{\mathrm{ts}}$ extracted by the infrared encoders, we perform spatio-temporal semantic fusion to enhance the representation of TC dynamics. First, a linear transformation layer, denoted as **Linear**, is applied to extract features corresponding to TC-related correlation factors. To effectively integrate these with the infrared features, we adopt a Self-Attention ConvLSTM [48], which is well-suited for capturing global spatio-temporal dependencies. This module enables the model to retain and propagate both spatial and temporal context across frames, thereby enriching the fused representation with deeper semantic information. The overall fusion process can be summarized as follows:

$$\mathbf{F}_{cor} = \mathbf{Linear}(\mathbf{X}_{\mathrm{cor}}; \theta_{\mathrm{cor}})$$
$$\mathbf{F}_{\mathrm{emb}} = \mathbf{Fus}_{\mathrm{sts}}(\mathcal{P}_{\mathrm{cat}}(\mathbf{F}_{\mathrm{ts}}^{\mathrm{i}}, \mathbf{F}_{\mathrm{cor}}); \theta_{\mathrm{sts}}) \tag{11}$$

where $\mathbf{Fus}_{\mathrm{sts}}$ denotes the ConvLSTM with Self-Attention and $\theta_{\mathrm{sts}}$ represents its parameters. Semantic fusion is achieved by introducing physical features at each step of temporal fusion through concatenation. Here, $\mathbf{F}_{\mathrm{ts}}^{\mathrm{i}} \in \{\mathbf{F}_{\mathrm{ts}}^{\mathrm{t_0}}, \mathbf{F}_{\mathrm{ts}}^{\mathrm{t_1}}\}$, and $\mathbf{F}_{\mathrm{emb}} \in \mathbb{R}^{c \times n}$, where $n$ is the feature dimension.

**Gaussian Distribution Sampling.** Based on semantic features after spatio-temporal semantic Fusion, good initial distribution is generated by Gaussian sampling. Specific sampling operations are as follows:

$$\boldsymbol{\mu} = \mathcal{P}_{\mathrm{m}}(\mathbf{F}_{\mathrm{emb}}, \mathbf{d}_1); \boldsymbol{\sigma} = \mathcal{P}_{\mathrm{s}}(\mathbf{F}_{\mathrm{emb}}, \mathbf{d}_1)$$
$$\mathbf{id} = \mathcal{P}_{\mathrm{gaussian}}(\boldsymbol{\mu}, \boldsymbol{\sigma}) \tag{12}$$

where $\mathcal{P}_{\mathrm{m}}$ and $\mathcal{P}_{\mathrm{s}}$ are operations of taking the mean and variance, the subscript of $\mathbf{d}_{\mathrm{i}}$ represents a specific dimension. $\mathcal{P}_{\mathrm{gaussian}}$ denotes a Gaussian sampling operation, which draws samples from a normal distribution using the mean and variance of the feature embeddings. In implementation, this corresponds to the torch.normal function in PyTorch.

## B.2 Task Dependency Flow Learning

**Holland Model.** The Holland model is a widely used empirical model in meteorology for representing the wind field of TCs [15], which can be used for estimating the wind speed profile of TC. The wind profile in the Holland model is assumed to decrease radially outward from the center of the TC. The model assumes a smooth, continuous decrease in wind speed with increasing distance from the center.

**Proof of Formula.** To incorporate the prior model for learning task identity, we make the following identical transformation to the original Holland equation.

$$\boldsymbol{r}^B \ln[(p_{\mathrm{n}} - p_{\mathrm{c}})/(p_{\mathrm{r}} - p_{\mathrm{c}})] = A$$
$$\Longleftrightarrow \boldsymbol{r}^B = \frac{A}{\ln[(p_{\mathrm{n}} - p_{\mathrm{c}})/(p_{\mathrm{r}} - p_{\mathrm{c}})]} = \gamma$$
$$\Longleftrightarrow \ln \boldsymbol{r}^B = \ln \gamma \tag{13}$$
$$\Longleftrightarrow B \ln \boldsymbol{r} = \ln \gamma$$
$$\Longleftrightarrow \boldsymbol{r} = \gamma^{\frac{1}{B}}$$

Based on the Holland model, we substitute the inner and outer-core size into Eq. (13), as shown below:

$$\mathtt{ro} = \gamma_{\mathtt{ro}}^{\mathtt{pow_{ro}}}, \gamma_{\mathtt{ro}} = \frac{A}{\ln[(p_{\mathrm{n}} - p_{\mathrm{c}})/(p_{\mathrm{ro}} - p_{\mathrm{c}})]}$$
$$\mathtt{ri} = \gamma_{\mathtt{ri}}^{\mathtt{pow_{ri}}}, \gamma_{\mathtt{ri}} = \frac{A}{\ln[(p_{\mathrm{n}} - p_{\mathrm{c}})/(p_{\mathrm{ri}} - p_{\mathrm{c}})]} \tag{14}$$

**PriorApp for TC Wind Speed.** Corresponding to Eq. (14), the prior constraint from wind speed to inner-core size is formulated as:

$$\gamma_{\mathtt{ri}} = \frac{\alpha_{\mathtt{v}}^{\mathtt{ri}}}{\ln\left(\mathcal{F}_{\mathtt{n}}(\mathbf{id}_{\mathtt{sp}}^{\mathtt{v}};\theta_{\mathtt{n}})\right) - \ln\left(\mathcal{F}_{\mathtt{r}}(\mathbf{id}_{\mathtt{sp}}^{\mathtt{v}};\theta_{\mathtt{r}})\right)}$$

$$\mathbf{U} = \mathrm{GConv}(([\mathbf{U}_1, \mathbf{U}_2, \ldots, \mathbf{U}_n], \mathbf{E}_{\mathtt{u}}); \theta_{\mathtt{u}})$$

$$\mathbf{id}_{\mathtt{sp}}^{\mathtt{ri}} = \mathrm{Iteration}^{\mathtt{t}}(\mathbf{H}^{\mathtt{t-1}}, \mathbf{U}, \gamma_{\mathtt{ro}}; \theta_{\mathtt{iter}}) = \mathbf{H}^{\mathtt{t}}, \text{ s.t. } \delta(\mathbf{H}^{\mathtt{t}}, \mathbf{H}^{\mathtt{t-1}}) < \tau \tag{15}$$

where $\ln(\cdot)$ denotes the natural logarithm; $\mathcal{F}_{\mathtt{n}}$ and $\mathcal{F}_{\mathtt{r}}$ are linear layers with learnable parameters $\theta_{\mathtt{n}}$ and $\theta_{\mathtt{r}}$; and $\alpha_{\mathtt{p}}^{\mathtt{ro}}$ is a learnable scaling factor. $\mathbf{U}_{\mathtt{i}}$ represents the learnable node embeddings, and $\mathbf{E}_{\mathtt{u}}$ is their edge connectivity matrix, where an edge exists if two nodes are adjacent. The GConv here refer to the Graph Convolutional Layer, which is introduced to supplement the prior equation by modeling the additional latent interactions through $\mathbf{U}$. The deep iterative algorithm terminates either upon reaching a predefined maximum number of iterations or when the difference $\delta(\mathbf{H}^{\mathtt{t}}, \mathbf{H}^{\mathtt{t-1}})$ falls below a threshold $\tau$.

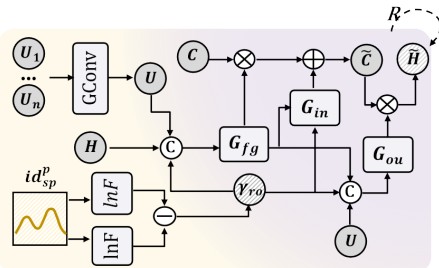

Figure 8: The framework of the proposed PriorAP.

**Details of PriorApp.** Inspired by the gating mechanisms in the Long Short-Term Memory (LSTM) architecture [43], we design a gated iterative module to approximate the power operation in a deep and dynamic manner. Specifically, we introduce a forget gate ($\mathcal{F}_{\mathtt{fg}}$), an input gate ($\mathcal{F}_{\mathtt{in}}$), and an output gate ($\mathcal{F}_{\mathtt{ou}}$), which collaboratively regulate the information flow across iterations, analogous to the memory cell updates in LSTM. As illustrated in Figure 8, the module refines its internal state over a maximum of $R$ iterations, or until convergence is reached, which is defined as the output fluctuation falling below a predefined threshold $\tau$. At each step, all relevant nodes are jointly considered as inputs for associative modeling. This iterative simulation allows the module to capture higher-order interdependencies among tasks, going beyond the expressiveness of the original power operation formulation. The internal states are updated at each iteration as follows:

$$\mathbf{f}_{\mathtt{fg}} = \sigma(\mathcal{F}_{\mathtt{fg}}([\mathbf{H}, \gamma_{\mathtt{ro}}, \mathbf{U}]; \theta_{\mathtt{fg}}))$$

$$\mathbf{f}_{\mathtt{in}} = \sigma(\mathcal{F}_{\mathtt{in}}([\mathbf{H}, \gamma_{\mathtt{ro}}]; \theta_{\mathtt{in}})) \cdot \tanh(\mathcal{F}_{\mathtt{c}}([\mathbf{H}, \gamma_{\mathtt{ro}}]; \theta_{\mathtt{c}}))$$

$$\mathbf{f}_{\mathtt{ou}} = \sigma(\mathcal{F}_{\mathtt{ou}}([\mathbf{H}, \gamma_{\mathtt{ro}}, \mathbf{U}]; \theta_{\mathtt{ou}}))$$

$$\widetilde{\mathbf{C}} = \mathbf{C} \cdot \mathbf{f}_{\mathtt{fg}} + \mathbf{f}_{\mathtt{in}} \,;\; \widetilde{\mathbf{H}} = \tanh(\widetilde{\mathbf{C}}) \cdot \mathbf{f}_{\mathtt{ou}}$$

$$\mathbf{id}_{\mathtt{sp}}^{\mathtt{ro}} = \widetilde{\mathbf{H}} \quad \text{s.t.} \quad \delta(\mathbf{H}) < \tau \tag{16}$$

Here, $\mathbf{H}$ and $\mathbf{C}$ denote the hidden state and cell state, respectively, both of which are iteratively updated via gated operations. The forget gate $\mathbf{f}_{\mathtt{fg}}$ determines how much of the previous memory $\mathbf{C}$ should be retained. The input gate $\mathbf{f}_{\mathtt{in}}$ incorporates new information into the memory by modulating a candidate state. The output gate $\mathbf{f}_{\mathtt{ou}}$ controls the extent to which the updated cell state is propagated to the output. The functions $\mathcal{F}_{\mathtt{fg}}, \mathcal{F}_{\mathtt{in}}, \mathcal{F}_{\mathtt{ou}},$ and $\mathcal{F}_{\mathtt{c}}$ are linear transformations, each parameterized by learnable weights $\theta_{\mathtt{fg}}, \theta_{\mathtt{in}}, \theta_{\mathtt{ou}},$ and $\theta_{\mathtt{c}}$, respectively. The functions $\sigma(\cdot)$ and $\tanh(\cdot)$ denote the sigmoid and hyperbolic tangent activations, respectively. The iterative process terminates when the change in the hidden state, measured by $\delta(\mathbf{H})$, falls below a predefined threshold $\tau$.

### B.3 Correlation-Aware Information Bridge

**Dark Knowledge Graph.** The graph $\mathbf{G}^{\mathtt{dk}}$ comprises node features extracted from $\mathbf{X}\mathtt{cor}$ and an edge index matrix defining connections between nodes. An edge is established between nodes i and j if $\mathbf{M}_{\mathtt{ij}}^{\mathtt{dk}} = 1$. Specifically, $\mathbf{M}^{\mathtt{dk}}$ refer to a correlation matrix, where the nodes correspond to correlation factors and TC attributes. An entry $\mathbf{M}_{\mathtt{ij}}^{\mathtt{dk}} = 1$ indicates that the $i$-th correlation factor is related to the $j$-th attribute. For example, the factor `tcf` (TC fullness) is related to TC size attributes such as `ri` and `ro`, resulting in $\mathrm{M}02^{\mathtt{dk}} = \mathrm{M}03^{\mathtt{dk}} = 1$. Based on prior physical knowledge of TC correlation factors, the initial matrix $\mathbf{M}^{\mathtt{dk}}$ is defined as follows:

$$
\mathbf{M}^{\text{dk}} = \begin{array}{c} \\ \texttt{tcf} \\ \texttt{tcc} \\ \texttt{tce} \\ \texttt{tcw} \end{array} \begin{array}{cccc} \texttt{v} & \texttt{p} & \texttt{ri} & \texttt{ro} \\ \left[ \begin{array}{cccc} 0 & 0 & 1 & 1 \\ 1 & 0 & 1 & 0 \\ 1 & 1 & 0 & 0 \\ 0 & 1 & 0 & 1 \end{array} \right] \end{array}
$$

Here, $\texttt{tcf}$, $\texttt{tcc}$, $\texttt{tce}$, and $\texttt{tcw}$ denote TC fullness, concentration ratio, energy ratio, and TC width, respectively. To address covariate and label shifts in both the input and output domains, $\mathbf{M}^{\text{dk}}$ is then used to construct the dark knowledge graph. Through the learning process, this graph captures latent physical correlations and leverages them to bridge the input–output gap via a task-shared identity token, denoted as $\mathbf{id}_{\text{sh}}$.

## C  Experiments

### C.1  Experiment Settings.

**Datasets.** To evaluate the performance of the proposed model in TC estimation, we use the Physical Dynamic TC datasets and the Digital TC datasets [46]. For constructing the Physical Dynamic TC datasets, we collected rich meteorological data from the IBTrACS [49] and satellite data from Himawari 8 [50] of TCs from the year 2015 and 2023. Since the infrared data of channels 7, 8, 13 and 15 are beneficial to TC estimation [2], we downloaded the brightness temperature data from these four IR channels which are normally used to monitor clouds and water vapor. The observation area is 60S–60N, 80E–160W, providing data at temporal and spatial resolutions of 10 min and 5 km, respectively. The satellite data were first linearly transformed to the interval [0, 1] using Min-Max Normalization. Then, the data were divided by year, with training data from 2015 to 2021. The training data were augmented by rotating the images by $90°$, $180°$, and $270°$ clockwise and flipping them horizontally and vertically. The data for 2022 and 2023 were used for validation and testing. Table 4 summarizes the number of samples in the training, validation, and test sets.

Table 4: Number of training, validation and test data.

| Dataset | Train | Valid | Test | Total |
|---|---|---|---|---|
| TC Num | 235 | 36 | 32 | 303 |
| File Num | 10099 | 1244 | 1245 | 12588 |

Moreover, to help model capture information of TC physical development, we matched the time dimension to obtain the developmental and correlation factors, including the category, the time since the TC became a named storm in minutes, the fullness, concentration ratio, energy ratio and width of TC. Among them, TC categories include tropical storm and hurricanes of grades 1 to 5.

For Digital TC datasets, we use the same ground-truth data set, i.e. International Best Track Archive for Climate Stewardship (IBTrACS) [49] to match it, and crop the satellite data size to $156 \times 156$ too. What's more, to conduct migration experiments on TC multi-task forecasting, we use the datasets used in MGTCF and TC-Diffuser [51, 52], which encompassing all the 1722 TCs data from 1950 to 2021 over the Western North Pacific.

**Experiment Settings.** To estimate multiple attributes of TC, we implemented the IDOL and all the State-Of-The-Art (SOTA) methods with the Pytorch toolkit on an NVIDIA RTX A6000 GPU. In our experiments, we employ the Adam optimization algorithm with an initial learning rate of 0.0001. The model is trained for 200 epochs with a batch size of 48. In addition, for a fair comparison, all experiments are performed on the same datasets, which will be publicly released.

### C.2  Performance Comparison.

The inference speed on the test datasets and model size of different methods are summarized in Table 6, which shows that IDOL is comparable to previous methods in terms of both model size and inference speed. Moreover, to better understand the model complexity, we analyze the parameter sizes of the proposed identity modeling modules and the feature extraction backbone (two VGG13

Table 5: Comparisons of TC multi-task estimation with previous methods on the Digital TC dataset.

| Categories | Comparison Methods | Wind Speed | | | Pressure | | | Inner-Core Size | | | Outer-Core Size | | |
|---|---|---|---|---|---|---|---|---|---|---|---|---|---|
| | | MAE | RMSE | STD | MAE | RMSE | STD | MAE | RMSE | STD | MAE | RMSE | STD |
| Traditional | ADT | 11.2 | 14.2 | - | 8 | 10.2 | - | - | - | - | - | - | - |
| | MTCSWA | - | - | - | - | - | - | 11.7 | 18.2 | - | 26.4 | 33 | - |
| Multi-Modal Fusion | STIA | 9.79 | 12.39 | 7.59 | - | - | - | - | - | - | - | - | - |
| | NS | - | - | - | - | - | - | 10.09 | 15.52 | 11.79 | 33.48 | 42.53 | 26.23 |
| | TC-MTLNet | 13.14 | 16.7 | 10.31 | 11.5 | 14.04 | 8.1 | - | - | - | 35.61 | 44.34 | 26.42 |
| | DeepTCNet | 9.15 | 11.87 | 7.56 | 8.02 | 9.91 | 5.82 | 7.99 | 11.76 | 8.62 | 28.91 | 36.11 | 21.63 |
| | PeRCNN | 6.51 | 8.54 | 5.53 | 6.89 | 8.57 | 5.09 | 7.88 | 11.57 | 8.48 | 28.17 | 34.9 | 20.59 |
| Invariant Learning | IRM | 9.3 | 11.81 | 7.28 | 7.65 | 9.44 | 5.53 | 8.07 | 12.4 | 9.41 | 28.6 | 34.7 | 19.57 |
| | V-Rex | 9.54 | 12.13 | 7.49 | 8.23 | 10.03 | 5.75 | 8.13 | 12.4 | 9.37 | 28.6 | 34.3 | 19.03 |
| | SADE | 9.55 | 11.93 | 7.15 | 8.33 | 10.22 | 5.93 | 7.77 | 11.8 | 8.91 | 28.5 | 34.2 | 18.88 |
| | DirMixE | 9.24 | 11.8 | 7.33 | 7.54 | 9.31 | 5.47 | 7.83 | 11.7 | 8.65 | 27.1 | 32.73 | 18.36 |
| | **IDOL** | **5.82** | **7.45** | **4.66** | **6.08** | **7.54** | **4.45** | **7.07** | **10.81** | **8.17** | **24.74** | **30.33** | **17.55** |

networks) in IDOL. The parameter sizes are 9.78M and 266.11M, respectively. This indicates that the identity distribution-oriented learning module itself is lightweight. To further investigate the trade-off between accuracy and efficiency, we replaced the VGG13 backbone with a significantly simpler CNN architecture and retrained the model on the same dataset. The comparative results are presented in Table 7.

Table 6: Comparisons of model size and inference time.

| Methods | DeepTCNet | **IDOL(ours)** |
|---|---|---|
| Size(M) | 270.42 | 275.89 |
| Inference time(s) | 4.88 | 3.18 |

Table 7: Comparisons of model size and inference time with different feature extraction backbone in IDOL, including two VGG13 networks and two CNN.

| Model | Size(M) | Time(s) | Wind Speed | | Pressure | | Inner-Core Size | | Outer-Core Size | |
|---|---|---|---|---|---|---|---|---|---|---|
| | | | MAE | RMSE | MAE | RMSE | MAE | RMSE | MAE | RMSE |
| DeepTCNet | 270.42 | 4.88 | 8.84 | 11.76 | 8.13 | 10.42 | 8.09 | 13.71 | 25.86 | 33.49 |
| **VGG-IDOL** | 275.89 | 3.18 | **5.93** | **7.60** | **5.77** | **7.15** | 6.24 | 12.06 | 17.06 | **23.26** |
| CNN-IDOL | **12.88** | **2.52** | 6.32 | 8.27 | 6.08 | 7.62 | **6.00** | **10.75** | **16.79** | 23.66 |

In summary, as shown in the table, replacing the backbone with a lightweight CNN architecture has slight impact on overall estimation performance. This demonstrates that the proposed identity distribution-oriented physical-invariant learning framework is not only effective and lightweight, but also robust to changes in the feature extraction backbone, making it highly adaptable and scalable for practical deployment.

**TC Multi-task Estimation.** Table 5 reports the overall performance of all the methods on Digital Typhoon dataset [46], where the best is shown in bold. The experimental results show that the proposed IDOL has the best performance in all tasks, including estimation error and model stability, even on the dataset with poor estimation performance. This further prove that the observational bias and inductive bias are potential in physical invariant learning. Moreover, we present the estimation performance for each TC category on the test set in Figure 9 and Figure 10. The consistently lower mean absolute error (MAE) and standard deviation (STD) across different TC categories demonstrate that our method exhibits stronger generalization and more robust learning capability when handling previously unseen TCs.

**TC Multi-task Forecasting.** MGTCF [51] and TC-Diffuser [52] are state-of-the-art methods for TC multi-task forecasting, based on Generative Adversarial Networks (GANs) and diffusion models, respectively. Therefore, these two models were selected for migration experiments. By integrating task-shared and task-specific identity tokens, which are guided by prior physical knowledge, into their original architectures, we observe notable improvements in both prediction accuracy and stability, as summarized in Table 9. Moreover, to further compare the performance of our approach with a foundation weather model, we selected the publicly available Pangu model as a representative baseline.

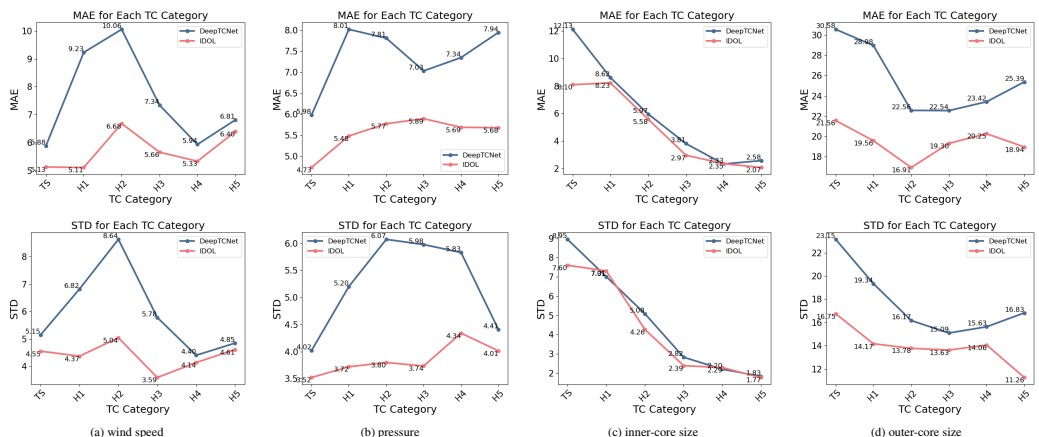

Figure 9: Visualization of estimation results across TC categories on our PDTC dataset. From left to right: estimated wind speed, pressure, inner-core size, and outer-core size.

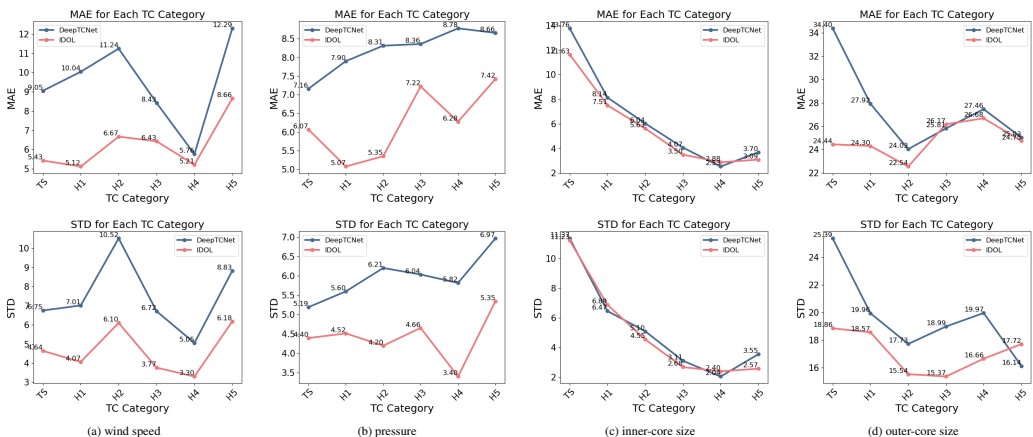

Figure 10: Visualization of estimation results across TC categories on the DigitalTC dataset. From left to right: estimated wind speed, pressure, inner-core size, and outer-core size.

To ensure fairness, we evaluated Pangu on the same datasets and reported its test results following the settings in TC-Diffuser [52]. These results demonstrate the effectiveness and generalizability of our physically invariant learning framework in the context of TC forecasting.

## C.3 Ablation Study.

To further validate the contribution of the Holland model in modeling task-specific identity tokens, we conducted additional experiments in which the Holland model was replaced with learnable linear layers or with intentionally perturbed priors (i.e. **Linear id$_{sp}$** and **Noisy id$_{sp}$**). Specifically, in the **Noisy id$_{sp}$** setting, we injected additional task correlations into the Holland model that are not physically supported in prior research, such as introducing an artificial dependency between inner-core size and outer-core size.

As shown in Table 8, the performance of the model with task-specific identities learned by Holland is better than that of the model with task-specific identities learned by Linear layers or noisy priors. It's because informing prior can help the model better capture the physical relationships among tasks and the development mechanism of tropical cyclones. Besides, from the table, we can also observe that the model with task-specific identities learned by Linear layers still performs better than the one without task-specific identity modeling, which demonstrates that even without prior knowledge, constraining the feature distribution to model task-specific identities is still effective for mitigating distribution shifts, thus improving the estimation accuracy of tropical cyclones.

Table 8: Additional Ablation experiments. $\mathbf{id}_{sp}$ represents task-specific and task-shared identity tokens.

| Settings | Wind Speed | | | Pressure | | | Inner-Core Size | | | Outer-Core Size | | |
|---|---|---|---|---|---|---|---|---|---|---|---|---|
| | MAE | RMSE | STD | MAE | RMSE | STD | MAE | RMSE | STD | MAE | RMSE | STD |
| Encoder Backbone | 10.13 | 13.25 | 8.54 | 7.79 | 10.13 | 6.47 | 8.32 | 13.87 | 11.1 | 28.58 | 37.66 | 24.52 |
| w/**Linear id**$_{sp}$ | 8.75 | 11.48 | 7.43 | 7.83 | 10.25 | 6.61 | 7.68 | 13.29 | 10.85 | 25.49 | 33.79 | 22.18 |
| w/ **Noisy id**$_{sp}$ | 8.54 | 11.33 | 7.44 | 7.58 | 9.88 | 6.34 | 8.1 | 13.95 | 11.36 | 25.84 | 33.84 | **21.84** |
| w/ **Holland id**$_{sp}$ | **7.24** | **9.13** | **5.55** | **6.66** | **8.27** | **4.97** | **7.37** | **13.24** | 10.99 | **24.91** | **33.28** | 22.07 |

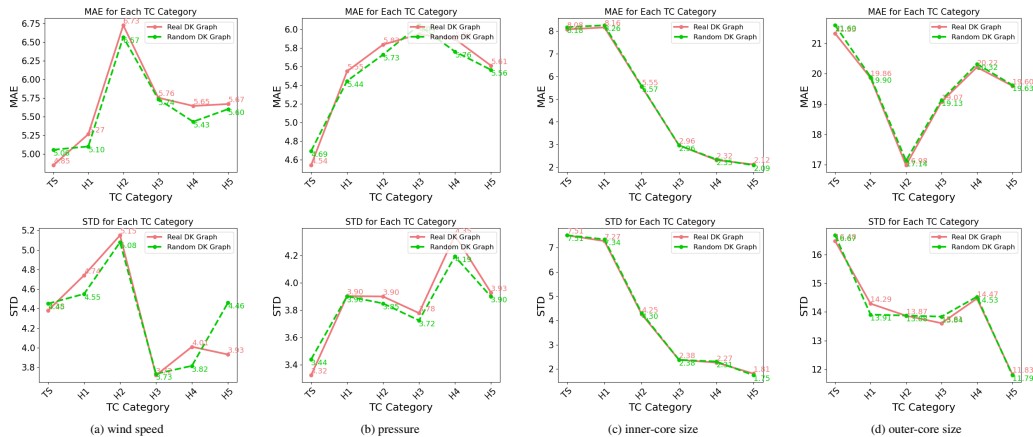

Figure 11: Visualization of estimation results using a random dark knowledge (DK) graph during testing. From left to right: estimated wind speed, pressure, inner-core size, and outer-core size.

## C.4 Analysis of Invariant learning

**Details of Shifted Sample Pairs.** To evaluate the model's robustness under distribution shifts, we screen out and visualize specific sample pairs from the test set that exhibit distribution shifts in both input and output. Following the task dependencies in the prior Holland model, pressure and outer-core size are treated as a correlated task group, with their shifted sample results shown in Figure 5. Three types of distribution shifts are illustrated in figure: covariate shift, referring to difference in TC structures as seen in the input; label shift, referring to distribution differences of the same output attribute across different TCs; and concept shift, referring to distribution differences across different output TC attributes. Similarly, the results for wind speed and inner-core size are presented in Figure 13.

To identify samples exhibiting covariate shift, we first compare the structural shapes and ground-truth values of TCs in the test dataset. A sample pair is considered to exhibit covariate shift if there is a substantial difference in TC structure (e.g., spatial pattern or appearance), while the difference in their GT values remains within a predefined threshold. Moreover, for detecting label shift and concept shift, we visualize the ground-truth distributions of each attribute across different TCs in the test set using kernel density estimation. A pair is considered to exhibit label shift if the distributions of the same attribute differ significantly across TCs. Similarly, a pair is considered to exhibit concept shift if the distributions of different TC attributes vary notably across instances. Specifically, the sample pairs exhibiting covariate shift in Figure 5 are drawn from the following TCs: {LAN-2023080918, HAIKUI-2023090318} and {LAN-2023081006, HAIKUI-2023090203}. In Figure 13, the covariate shift sample pairs are from {DAMREY-2023082718, KIROGI-2023083118} and {MAWAR-2023052112, MAWAR-2023053015}. Additional estimation results under covariate shift are shown in Figure 14, where the selected sample pairs are from the following TC instances: {HAIKUI-2023090212, LAN-2023081009}, {HAIKUI-2023090318, LAN-2023081000}, {LAN-2023080918, MAWAR-2023053015}, and {BOLAVEN-2023101306, KOINU-2023100300}. Additional results under label shift and concept shift are provided in Figure 15.

Table 9: Experimental results of method migration to verify the ability to handle distribution shifts in TC multi-task forecasting.

| Methods | Metrics | Trajectory (km) | | | | Pressure (hpa) | | | | Wind Speed (m/s) | | | |
|---|---|---|---|---|---|---|---|---|---|---|---|---|---|
| | | 6h | 12h | 18h | 24h | 6h | 12h | 18h | 24h | 6h | 12h | 18h | 24h |
| Pangu | MAE | 42.8 | 44.75 | 50.85 | 65.68 | 16 | 16.5 | 16.7 | 16.9 | - | - | - | - |
| MGTCF | MAE | 24.21 | 44.99 | 69.2 | 96.77 | 1.41 | 2.14 | 2.77 | 3.25 | 0.79 | 1.2 | 1.57 | 1.88 |
| MGTCF w/ IDOL | | **24.21** | **44.23** | **68.4** | **95.81** | **1.31** | **1.95** | **2.38** | **2.82** | **0.74** | **1.14** | **1.42** | **1.64** |
| MGTCF | STD | 20.31 | 36.79 | 55.27 | 81.87 | 2.25 | 3.01 | 3.81 | 4.35 | 1.17 | 1.58 | 1.89 | 2.14 |
| MGTCF w/ IDOL | | **19.48** | **33.99** | **49.77** | **73.99** | **1.96** | **2.52** | **3** | **3.4** | **1.04** | **1.42** | **1.69** | **1.85** |
| TC-Diffuser | MAE | 19.39 | 20.83 | 41.82 | 77.41 | 1.24 | 0.84 | 1.85 | 2.81 | 0.75 | 0.43 | 0.93 | 1.38 |
| TC-Diffuser w/ IDOL | | **18.73** | **19.53** | **40.82** | **74.21** | **1.21** | **0.81** | **1.66** | **2.47** | **0.73** | **0.37** | **0.91** | **1.31** |
| TC-Diffuser | STD | 21.16 | 25.15 | 44.47 | 73.78 | 1.83 | 1.35 | 2.88 | 4.31 | 0.96 | 0.7 | 1.46 | 2.21 |
| TC-Diffuser w/ IDOL | | **20.55** | **22.29** | **39.35** | **65.85** | **1.81** | **1.16** | **2.44** | **3.69** | 0.99 | **0.57** | **1.42** | **2.02** |

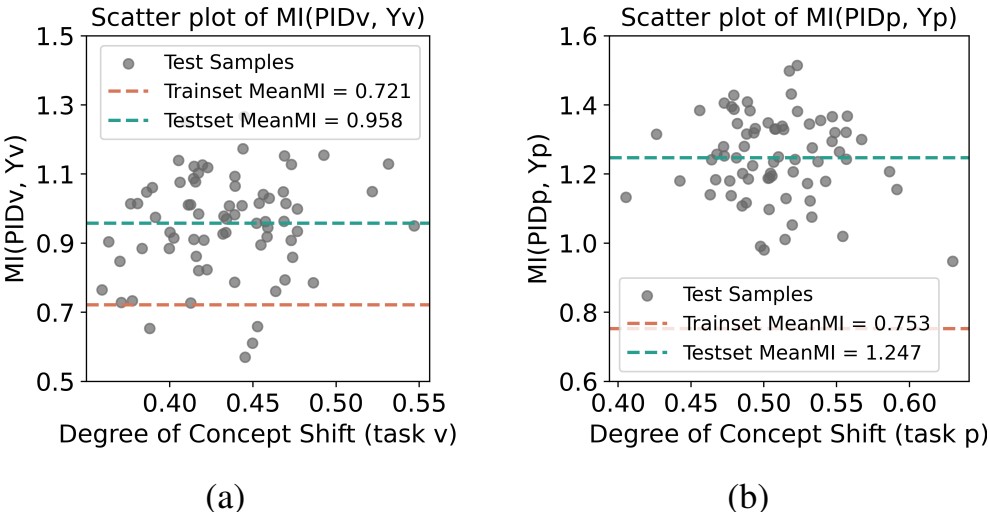

Figure 12: Validation of task dependency modeling for (a) $PID_v$-$Y_v$ and (b) $PID_p$-$Y_p$: Scatter plots of mutual information (MI) used to quantify the adequacy of correlation modeling.

**Performance of Invariant learning.** Compared with SOTA method DeepTCNet, in Figure 5(a) and 13(a), the estimations of our IDOL (green boxes with underlines) are noticeably closer to the ground truth (red boxes), even under covariate shift. Meanwhile, in Figure 5(b) and 13(b), IDOL also achieves lower estimation error across different TC instances and attributes with varying distributions, showing its effectiveness under label and concept shifts. These results collectively demonstrate IDOL's robustness to various distribution shifts, validating its capability in physical invariance learning.

### C.5 Analysis of Physical Identity

The visualizations and additional analysis results discussed in Subsection 5.4 are provided below:

- Figure 12: Scatter plot showing MI vs. concept shift magnitude.
- Figure 11: Comparison between real and random dark knowledge graphs on model performance (MAE and STD).

**Definition of Shift Degree.** To quantitatively measure the degree of distribution shift between the training and test sets, we define the *shift magnitude* using the **Jensen–Shannon divergence (JSD)** between the attribute distributions. Specifically, for each TC attributes, we estimate its empirical probability density on the training set and on the test set via kernel density estimation. The JSD

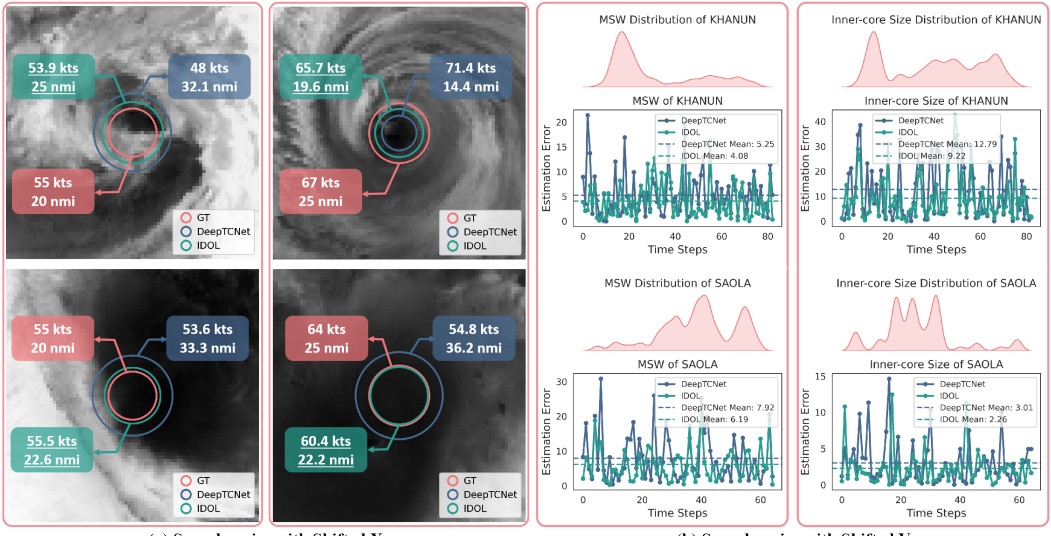

Figure 13: Estimation performance of wind speed and inner-core size on test samples under distribution shifts: (a) Estimation results under covariate shift and (b) Mean absolute errors under label and concept shifts across different TCs and output attributes. Each vertical red box indicates a pair of samples exhibiting distribution shift.

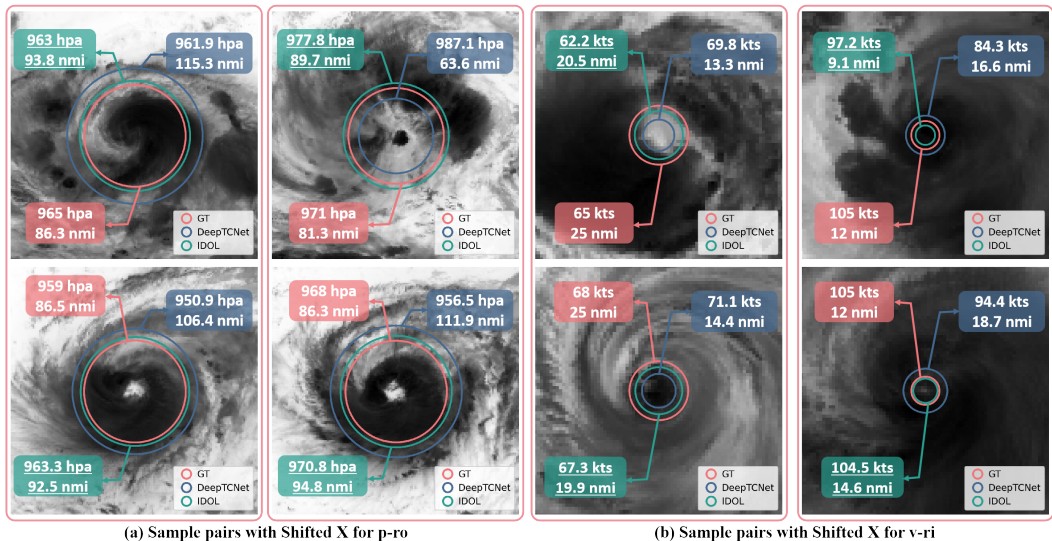

Figure 14: Estimation performance under covariate shift for (a) pressure and outer-core size (p-ro) and (b) wind speed and inner-core size (v-ri). Each vertical red box indicates a pair of samples exhibiting distribution shift.

between the two distributions is then computed and used as a scalar measure of the distribution shift:

$$\text{JSD}(P \parallel Q) = \frac{1}{2}\text{KL}(P \parallel M) + \frac{1}{2}\text{KL}(Q \parallel M), \quad \text{where } M = \frac{1}{2}(P + Q) \tag{17}$$

Here, $\text{KL}(\cdot \parallel \cdot)$ denotes the Kullback–Leibler divergence, and $P$, $Q$ are the estimated distributions for training and test sets, respectively. JSD is a symmetric and bounded measure, which provides a stable metric for quantifying both covariate and label shift degree.

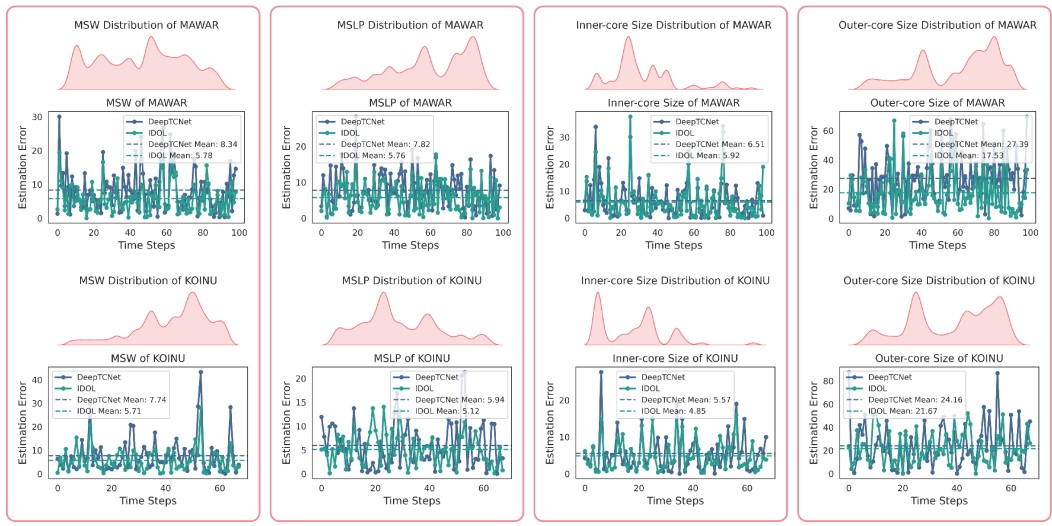

Figure 15: Estimation performance under label shift and concept shift. Each vertical red box indicates a pair of samples exhibiting distribution shift.

# D    Limitations and Future Work

Despite the promising performance of IDOL, our exploration of physical invariance in the embedding space remains at a preliminary stage. The prior wind field model (e.g., the Holland model) incorporated into the framework has inherent statistical simplifications and limitations. While the proposed correlation-aware information bridge helps compensate for physical correlations not captured by the Holland model, the overall robustness of the model can still be improved. Future work may explore constraining the feature space using more comprehensive and generalizable physical mechanisms, such as fundamental dynamical or thermodynamic equations. Incorporating such principles has the potential to further enhance model generalization under diverse and complex distribution shifts.

