# OpenReview forum: "IDOL: Meeting Diverse Distribution Shifts with Prior Physics for Tropical Cyclone Multi-Task Estimation"
_NeurIPS.cc/2025/Conference — NeurIPS 2025 poster_

### Official Review · Reviewer_SbdX · 2025-06-27

**Clarity:** 3
**Significance:** 3
**Originality:** 3
**Rating:** 4
**Confidence:** 3

**Summary:**

This work tackles distribution shifts (covariate, label, concept shifts) in tropical cyclone multi-task estimation by proposing IDOL, a physics-guided framework that imposes identity-oriented constraints on feature distributions. Leveraging the Holland wind field model and dark knowledge correlations, IDOL decomposes task dependencies into physical chains (e.g., wind speed → core sizes) via task-specific tokens and bridges input-output distributions via task-shared tokens. Extensive experiments  conducted on multiple datasets and tasks demonstrate the outperformance of the proposed IDOL.

**Questions:**

1. The task operates under a multi-task learning framework. The network structure involves shared and task-specific components. How do changes in the ratio of shared-to-task-specific components (e.g., the degree of parameter sharing) affect model size and performance metrics?"
2. Physical priors (Holland model) are central to IDOL but lack ablation on their necessity.

**Ethical Concerns:**

["NO or VERY MINOR ethics concerns only"]

**Limitations:**

yes

**Quality:**

3

**Strengths And Weaknesses:**

Strengths:
1. Novel Physics-Guided Framework. Proposes IDOL to address simultaneous distribution shifts (covariate/label/concept shifts) in TC estimation using physics-informed identity tokens. This bridges a critical gap between invariant learning and meteorological priors.
2. Technical Sophistication: Elegantly integrates the Holland wind field model (§4.1) and dark knowledge correlations (§4.2) to decompose task dependencies and constrain feature distributions.
3. Rigorous Experimental Validation.

Weaknesses
1. Lacks formal proofs for invariance guarantees.

---

> ### Author Rebuttal · Authors · 2025-07-29
>
> We are pleased that the reviewer recognized the novelty of our physics-guided framework and the rigor of our experimental validation. We also sincerely appreciate the constructive feedback regarding the experimental demonstrations, which have been instrumental in improving the quality of our manuscript. The reviewers' comments are shown below in italicized font, and our responses are presented in regular font. We hope that these answers will meet with your approval.
>
> **Comment 1**: *The task operates under a multi-task learning framework. The network structure involves shared and task-specific components. How do changes in the ratio of shared-to-task-specific components (e.g., the degree of parameter sharing) affect model size and performance metrics?"*
>
> **Response 1**: Thank you for your valuable comments. In our work, to address various distribution shifts, we propose IDOL, which characterizes the distribution of feature embeddings to model both task-shared and task-specific invariant identity tokens. **Unlike previous multi-task learning frameworks that typically separate shared and specific components from backbone features, our approach adopts physical priors to regulate the distributions of the features to learn physical invariance. Specifically, the original feature distributions are guided by physical priors and are then separately transformed into task-shared and task-specific identity tokens.**
>
> However, to explore how different degrees of parameter sharing affect model size and multi-task performance, we conducted supplementary experiments by varying the parameter ratios between task-shared and task-specific identity tokens. Specifically, we define the ratio based on the dimensional size of the corresponding identity token, with configurations including 1:1, 1:2, 1:3, 2:1, and 3:1. For example, a ratio of 1:2 means that the task-shared identity token has half the dimensionality of the task-specific identity token. The results are presented below.
>
> | Model           | Model Size (MB) | Wind MAE | Wind RMSE | Wind STD | Pressure MAE | Pressure RMSE | Pressure STD | Inner-Core MAE | Inner-Core RMSE | Inner-Core STD | Outer-Core MAE | Outer-Core RMSE | Outer-Core STD |
> |-----------------------------|------------------|---------|-----------|---------|----------|------------|-----------|----------|------------|----------|----------|------------|----------|
> | **IDOL (1:1)**              | **1124.31**          | 5.93 | 7.60 | **4.75** | **5.77** | **7.15**  | 4.23 | **6.24** | **12.06**     | **10.31** | **17.06** | **23.26** | **15.80** |
> | 1:2             | 1124.35          | **5.75**     | **7.46**      | 4.76    | 5.78     | 7.17       | 4.25     | 6.36     | 12.24 | 10.45     | 17.95     | 24.74     | 17.02     |
> | 1:3             | 1124.39          | 6.10     | 7.85      | 4.95    | 5.95     | 7.36       | 4.33     | 6.30     | 12.07 | 10.29     | 18.11     | 24.52     | 16.53     |
> | 2:1             | 1124.35          | 5.91 | 7.58      | 4.75    | 5.9     | 7.33       | 4.35 | 6.32     | 12.03     | 10.24     | 17.94     | 24.52     | 16.72     |
> | 3:1             | 1124.38          | 5.93 | 7.69      | 4.89    | 5.79     | 7.14       | **4.17** | 6.13     | 11.81     | 10.09     | 18.79     | 25.34     | 17.00     |
>
> **Note**: Wind, Inner-Core, and Outer-Core refer to the estimation tasks of wind speed (m/s), inner-core size (nmi), and outer-core size (nmi), respectively.
>
> **In summary, as shown in the table, under the given model design and experimental settings, varying the parameter ratio between task-shared and task-specific identity tokens has slight impact on model size and overall estimation performance. This is mainly because the proposed identity tokens, even with lightweight configurations, are sufficient to impose physical constraints on the feature distributions, thereby guiding the model to effectively capture the intrinsic characteristics required for multi-task learning.**
>
> Increasing the dimensionality of a specific identity token primarily enables more fine-grained representation learning, which may lead to slight variations in task-wise performance, but does not result in significant overall performance gains. This further confirms the effectiveness and efficiency of the proposed identity token design in ensuring consistent and robust multi-task learning.
>
> **Comment 2**: *Physical priors (Holland model) are central to IDOL but lack ablation on their necessity.*
>
> **Response 2**: Thank you for your thoughtful review. To further validate the contribution of the Holland model in modeling task-specific identities, we conducted supplementary experiments in which the priors is replaced with learnable linear layers. The corresponding results, which will be included in the revised manuscript, are presented as follows:
>
> | Method       | Wind MAE | Wind RMSE | Wind STD | Pressure MAE | Pressure RMSE | Pressure STD | Inner-Core MAE | Inner-Core RMSE | Inner-Core STD | Outer-Core MAE | Outer-Core RMSE | Outer-Core STD |
> |--------------|----------|-----------|----------|---------------|----------------|---------------|------------------|-------------------|----------------|------------------|-------------------|----------------|
> | VGG-LSTM (w/o $\mathbf{id}_\mathtt{sp}$)      |   10.13 |    13.25 |    8.54 |      7.79 |      10.13 |      6.47 |     8.32 |    13.87 |   11.09 |    28.58 |    37.65 |   24.52 |
> | w/ $\mathbf{id}_\mathtt{sp}$ by Holland   |    **7.24** |     **9.13** |    **5.55** |      **6.60** |       **8.27** |      **4.97** |     **7.37** |    **13.24** |   10.99 |    **24.91** |    **33.28** |   **22.07** |
> | w/ $\mathbf{id}_\mathtt{sp}$ by Linears    |    8.75 |    11.48 |    7.43 |      7.83 |      10.25 |      6.61 |     7.68 |    13.29 |   10.85 |    25.49 |    33.79 |   22.18 |
>
> **Note**: Wind, Inner-Core, and Outer-Core refer to estimation tasks of Wind Speed (m/s), Inner-Core Size (nmi), and Outer-Core Size (nmi) errors, respectively. $\mathbf{id}_\mathtt{sp}$ refers to the proposed task-specific identitiy tokens.
>
> **In summary, the performance of the model with task-specific identities modeled by Holland is better than that of the model with task-specific identities learned by Linear layers. It’s because informing prior can help the model better capture the physical relationships among tasks and the development mechanism of tropical cyclones. Besides, from the table, we can also observe that the model with task-specific identities learned by Linear layers still performs better than the one without task-specific identity modeling, which demonstrates that even without prior knowledge, constraining the feature distribution to model task-specific identities is still effective for mitigating distribution shifts, thus improving the estimation accuracy of tropical cyclones.**
>
> **Comment 3**: *Lacks formal proofs for invariance guarantees.*
>
> **Response 3**: Thank you for the valuable comment, which helps improve the theoretical completeness of our work. At this stage, the proposed IDOL is designed to learn invariant feature distributions in the tropical cyclone (TC) domain from a physical and causal modeling perspective. The learned invariance is empirically validated through comprehensive and rigorous experiments.
>
> We are actively exploring a more principled theoretical framework to provide formal invariance guarantees. However, the representation of distribution shifts arising from the complex and evolving nature of TCs remains highly abstract and challenging to formulate explicitly. In the future, this limitation may be addressed by further integrating Bayesian Invariant Learning and Physics-Informed Learning, which together can model uncertainty while ensuring physical consistency across diverse environments.

---

> ### Author Response · Authors · 2025-08-08
> **About the rebuttal**
>
> Dear Reviewer, we would like to confirm whether our rebuttal has sufficiently addressed your concerns or not. If there are any remaining questions or clarifications needed, we would be glad to discuss them further at this stage. Thank you in advance.

---

### Official Review · Reviewer_3jY9 · 2025-06-29

**Clarity:** 3
**Significance:** 2
**Originality:** 3
**Rating:** 4
**Confidence:** 4

**Summary:**

The paper presents Identity Distribution-Oriented Physical Invariant Learning framework (IDOL), which is a learning method that addresses different types of distribution shifts for tropical cyclone multi-task estimation.  The framework is diverse enough to handle label and semantic shifts.  The method utilizes proven data-centric approaches and presents comprehensive evaluation.

**Questions:**

-Is there a way to show visually the split between invariant and variant components?
-These phenomena change with changing atmospheric and other dynamics. Is there an efficient way to infuse new external variables?
-How can this framework work in an unsupervised way?

**Ethical Concerns:**

["NO or VERY MINOR ethics concerns only"]

**Limitations:**

Yes.

**Paper Formatting Concerns:**

None.

**Quality:**

3

**Strengths And Weaknesses:**

Strengths:
-The architecture is well thought and addresses invariant and variant features.
-The application of the framework has real-world implications.
-Comprehensive evaluation shows improvements over existing methods.

Weaknesses:
-The trade-offs between slight performance over the new architecture are not explained.
-It is unclear how much ground truth labels (which may not be available) are required for addressing domain shifts.
-The paper can be strengthened with some considerations on simplification of the architecture to reduce the deployment since this has real word implications.

---

> ### Author Rebuttal · Authors · 2025-07-29
>
> We are pleased that the reviewer recognized the well-designed architecture and comprehensive evaluations of our work. We also sincerely appreciate the constructive comments on the contribution demonstration, which have been highly valuable in helping us improve the manuscript. The reviewers' comments are shown below in italicized font, and our responses are presented in regular font. We hope that these answers will meet with your approval.
>
> **Comment 1**: *The trade-offs between slight performance over the new architecture are not explained. -It is unclear how much ground truth labels (which may not be available) are required for addressing domain shifts. -The paper can be strengthened with some considerations on simplification of the architecture to reduce the deployment since this has real word implications.*
>
> **Response 1**: Thank you for the valuable comments. As demonstrated in Table 1 of the main text and Table 4 in Appendix C.2, our proposed method achieves the best estimation performance without increasing model parameters or inference time. Since all multi-attribute labels for tropical cyclones are publicly available, our approach is implemented within a supervised learning framework. By jointly considering the short inference time and low computational complexity of stacked convolutions, IDOL achieves the balance between prediction accuracy and computational efficiency, making it readily deployable in real-world applications.
>
> Moreover, to better understand the model complexity, we analyze the parameter size of the proposed identity modeling modules and the feature extraction backbone (two VGG13 networks) in IDOL. The parameter size of them are 9.78M and 266.11M, respectively. This indicates that the identity distribution-oriented learning module itself is lightweight. To further investigate the trade-off between accuracy and efficiency, we replaced the VGG13 backbone with a significantly simpler CNN architecture and retrained the model on the same dataset. The comparative results are presented below:
> | Model           | Parameter Size (Million) | Inference Time (s)  | Wind MAE | Wind RMSE | Wind STD | Press MAE | Press RMSE | Press STD | Inner-Core MAE | Inner-Core RMSE | Inner-Core STD | Outer-Core MAE | Outer-Core RMSE | Outer-Core STD |
> |-----------------------------|---------|---------|---------|-----------|---------|----------|------------|-----------|----------|------------|----------|----------|------------|----------|
> | DeepTCNet    | 270.42  | 4.88  | 8.84     | 11.76     | 7.76     | 8.13          | 10.42          | 5.31          | 8.09             | 13.71             | 11.25          | 25.86            | 33.49             | 21.29          |
> | ***VGG w/ IDOL***     | *275.89*   | *3.18*    | ***5.93*** | ***7.60*** | ***4.75*** | ***5.77*** | ***7.15***  | ***4.23*** | *6.24* | *12.06*     | *10.31* | *17.06* | ***23.26*** | ***15.80*** |
> | *CNN w/ IDOL*       | ***12.88***     | ***2.52***   | *6.32*     | *8.27*      | *5.34*    | *6.08*     | *7.62*       | *4.59*     |  ***6.00***     | ***10.75*** |  ***8.92***     |  ***16.79***     | *23.66*     | *16.67*     |
>
> **Note**: Wind, Press, Inner-Core, and Outer-Core refer to the estimation tasks of wind speed (m/s), pressure (hpa), inner-core size (nmi), and outer-core size (nmi), respectively.
>
> **In summary, as shown in the table, replacing the backbone with a lightweight CNN architecture has slight impact on the performance on overall estimation performance. This demonstrates that the proposed identity distribution-oriented physical-invariant learning framework is not only effective and lightweight, but also robust to changes in the feature extraction backbone, making it highly adaptable and scalable for practical deployment.**
>
> **Comment 2**: *Is there a way to show visually the split between invariant and variant components?*
>
> **Response 2**: Yes, it is possible to visualize the separation between invariant and variant components by first applying dimensionality reduction on them and then using kernel density estimation (KDE) to show visually their distributions across different time domains. However, as the rebuttal box does not support figure attachments, we do not present the result here, which will be added in the revised appendix.
>
> Additionally, as shown in Figure 4(b) of the manuscript, we provide an intuitive illustration of the invariant and variant components across different temporal domains. In the context of invariant learning, features from different domains should remain close in distribution, which implies similar statistical variances across domains. Taking the components shown in Figure 4(b) as an example, the task-shared identity ($PID_{sh}$) and task-specific identities ($PID_v$, $PID_p$, $PID_{ri}$, and $PID_{ro}$) consistently exhibit low variance across time domains, and are therefore regarded as invariant components. In contrast, the features extracted by the baseline DeepTCNet demonstrate fluctuating variance across different temporal domains, which are regarded as variant components.
>
> **Comment 3**: *These phenomena change with changing atmospheric and other dynamics. Is there an efficient way to infuse new external variables?*
>
> **Response 3**: Yes, there is an efficient way to incorporate new external variables. Since the proposed Correlation-Aware Information Bridge consists of graph convolutions and learnable parameters of a mixed Gaussian distribution, we can effectively infuse new external environmental variables by increasing the number of nodes in the knowledge graph. This enables the model to better learn representations of input data and their latent correlations with output tasks, while applying these effects to constrain feature distributions.
>
> **Comment 4**: *How can this framework work in an unsupervised way?*
>
> **Response 4**: Thank you for your insightful question. We believe our framework can be extended to an unsupervised setting in future work. One potential direction is to perform clustering on the TC convection structures to extract inherent patterns without label supervision. Another promising approach is to leverage contrastive learning between satellite images and descriptive text to align multi-modal features in an unsupervised manner. These adaptations would allow the model to learn meaningful representations without relying on explicit labels.

---

> ### Author Response · Authors · 2025-08-08
> **About the rebuttal**
>
> Dear Reviewer,  we would like to confirm whether our rebuttal has sufficiently addressed your concerns or not. If there are any remaining questions or clarifications needed, we would be glad to discuss them further at this stage. Thank you in advance.

---

### Official Review · Reviewer_gmF3 · 2025-07-02

**Clarity:** 3
**Significance:** 3
**Originality:** 3
**Rating:** 4
**Confidence:** 2

**Summary:**

The goal of this paper is to explore prior physical knowledge for improving out of distribution generalization in problems of tropical cyclone estimation. The key contribution is merging physics with machine learning to handle distribution shifts. Evaluation is done on diverse datasets.

**Questions:**

- Can you show more dynamic use cases in your experiments? In other words, can you simulate for highly changing distribution shifts?
- Can the proposed approach handle domain shifts?
- What happens if the model priors are erroneous and/or approximate?
- How does this approach compare to other invariant representation learning techniques?

**Ethical Concerns:**

["NO or VERY MINOR ethics concerns only"]

**Final Justification:**

The authors did a good job in the revision, I think this paper could be accepted if there is room

**Limitations:**

Limitations are properly discussed

**Quality:**

3

**Strengths And Weaknesses:**

Strengths:
+ The use of physical priors to improve out-of-distribution (OOD) generalization is timely and important.
+ The focus on a specific tropical cyclone use case properly grounds the research.
+ The experimental results are promising.

Weaknesses:
- It is not clear what happens to the performance is the prior models are approximate or not fully accurate.
- For the cases of tropical cyclones, it is not clear how often OOD issues must be handled. The scenario does not see overly dynamic.

---

> ### Author Rebuttal · Authors · 2025-07-29
>
> We are pleased that the reviewer recognized the effectiveness of our prior incorporation strategy and the promising experimental results. Moreover, we sincerely appreciate the valuable suggestions regarding experimental demonstrations, which have greatly helped us to revise and improve our paper. The reviewers' comments are shown below in italicized font, and our responses are presented in regular font. We hope that these answers will meet with your approval.
>
> **Comment 1**: *(weakness) It is not clear what happens to the performance if the prior models are approximate or not fully accurate.*  & *(question) What happens if the model priors are erroneous and/or approximate?*
>
> **Response 1**: Thank you for the insightful comment. In our approach, we propose the Prior-aware Approximator (PriorAPP) to incorporate the Holland model as prior knowledge. Since Holland is a classical statistical model in tropical cyclone (TC) research, it is well known that it is not fully accurate. As shown in the first two rows of Table 2 in the manuscript, the task-specific identities ($\mathbf{id}_\mathtt{sp}$) are derived based on the Holland model. Therefore, the second row of Table 2 in the manuscript reflects the model’s estimation performance when the prior is not fully accurate. The superior performance of the model with Holland demonstrates both the effectiveness of incorporating physical priors and the robustness of our method to approximate priors.
>
> As shown in the following table, to further validate the contribution of the prior model and the robustness of our design, we conducted additional experiments under intentionally perturbed prior conditions. Specifically, in the "*w/ Noisy Prior*" setting, we inject additional task correlations into the Holland model that are not physically validated in previous research, such as introducing an artificial dependency between inner-core size and outer-core size.
>
> | Method       | Wind MAE | Wind RMSE | Wind STD | Pressure MAE | Pressure RMSE | Pressure STD | Inner-Core MAE | Inner-Core RMSE | Inner-Core STD | Outer-Core MAE | Outer-Core RMSE | Outer-Core STD |
> |--------------|----------|-----------|----------|---------------|----------------|---------------|------------------|-------------------|----------------|------------------|-------------------|----------------|
> | VGG-LSTM           |   10.13 |    13.25 |    8.54 |      7.79 |      10.13 |      6.47 |     8.32 |    13.87 |   11.09 |    28.58 |    37.65 |   24.52 |
> | w/ Prior |    **7.24**|     **9.13** |    **5.55** |      **6.60** |       **8.27** |      **4.97** |     **7.37** |    **13.24** |   **10.99** |    **24.91** |    **33.28** |   **22.07** |
> | *w/ Noisy Prior*   |    *8.54*|     *11.33* |    *7.44* |      *7.58* |       *9.88* |      *6.34* |     *8.1* |    *13.95* |   *11.36*|    *25.84* |    *33.84* |   *21.84* |
>
> **Note**: Wind, Inner-Core, and Outer-Core refer to estimation tasks of Wind Speed (m/s), Inner-Core Size (nmi), and Outer-Core Size (nmi) errors, respectively.
>
> **These results clearly indicate that the model maintains stable and robust performance even under perturbed prior conditions. This suggests that the proposed Prior-aware Approximator, combined with the nonlinear representation power of deep networks, can effectively adapt to approximate and erroneous priors. These supplementary experimental results will be added to the appendix in the revised version.**
>
> **Comment 2**: *For the cases of tropical cyclones, it is not clear how often OOD issues must be handled. The scenario does not see overly dynamic.*
>
> **Response 2**: Thank you for the insightful comment. To assess the extent of distributional shifts in tropical cyclone (TC) estimation, we analyzed the label distributions across all TCs in the training and test sets. Specifically, we compute the Jensen-Shannon Divergence (JSD) between each test TC’s label distribution and that of the training set. A higher JSD indicates a more significant distribution shift. Based on median statistics, we empirically set a threshold of 0.39 to define out-of-distribution (OOD) cases. Our results show that 44% of test samples exceed this threshold, confirming that OOD issues represent a significant challenge in tropical cyclone estimation that warrants further attention and dedicated handling.
>
> Moreover, from a meteorological perspective, the development of tropical cyclone (TC) attributes is inherently dynamic and nonlinear. Their evolution is influenced by internal processes such as eyewall replacement cycles, convective bursts, and variations in structural features like intensity distribution and inner-core compactness [1]. These dynamic behaviors introduce significant variability in TC structure and intensity evolution. As a result, no training set can fully capture the entire spectrum of possible TC states, highlighting the necessity of building models with strong generalization ability under distribution shifts. This is precisely the motivation behind our proposed identity distribution-oriented learning framework.
> - [1] Rojas, B., Didlake, A., & Zhang, J. (2024). Asymmetries During Eyewall Replacement Cycles of Hurricane Ivan (2004). Monthly Weather Review.
>
> **Comment 3**: *Can you show more dynamic use cases in your experiments? In other words, can you simulate for highly changing distribution shifts?*
>
> **Response 3**: Yes, we can show more dynamic use cases in our experiments. However, due to the inability to attach pictures in the rebuttal box, these cannot be displayed here. Nevertheless, as demonstrated in Figure 5 of the main text and Figures 11 to 13 in Appendix C.2, the presented cases already exhibit significant distribution shifts, effectively showing IDOL's capability to handle distribution shifts. Specifically, Figure 5(a) in the main text, Figure 11(a) and Figure 12 in Appendix C.2 highlight sample pairs with distinct TC shape organizations, e.g., a blurred center versus a clear eye shape, which indicates significant input data distribution shifts. Additionally, Figure 5(b) in the main text, Figure 11(b) and Figure 13 in Appendix C.2 illustrate notable label distribution shifts, reflected in entirely different distribution patterns.
>
> **Comment 4**: *Can the proposed approach handle domain shifts?*
>
> **Response 4**: Yes, we believe the proposed approach can handle domain shifts for the following reasons:
>
> Firstly, researches on domain shifts often employ domain-invariant feature learning, using visualizations of stable feature distributions across different domains to demonstrate their methods' effectiveness [2, 3]. This suggests that domain shifts can be partially reflected through feature distribution shifts. Our approach leverages prior knowledge to constrain feature distributions with physical invariance, aligning the goal of modeling invariant task identities with extracting domain-invariant features. The key distinction of our IDOL lies in integrating physical priors into distribution constraints, which we argue enhances generalization learning capability.
>
> Secondly, in invariant learning, features from different domains should be close to each other, which means they will have similar variances. Given that tropical cyclones across different years exhibit unique developmental traits and variable environments, temporal domain shifts are evident. If our model extracts physically invariant identity tokens across these time domains, their statistical variance should remain consistent. As shown in Figure 4(b) of the manuscript, our physical identity tokens display smaller variance across time domains compared to DeepTCNet features, highlighting greater robustness to domain shifts. This improvement supports that incorporating prior physics enables the model to better understand the internal TC dynamics and generalize to unseen, variable environments.
> - [2] Chen, Yang, et al. "Achieving domain generalization for underwater object detection by domain mixup and contrastive learning." Neurocomputing 528 (2023): 20-34.
> - [3] Chen, Haojie, et al. "Self-supervised domain feature mining for underwater domain generalization object detection." Expert Systems with Applications 265 (2025): 126023.
>
> **Comment 5**: *How does this approach compare to other invariant representation learning techniques?*
>
> **Response 5**: In Table 1 of the manuscript, we have already compared the proposed IDOL with other invariant representation learning methods, including classic approaches (e.g., IRM, V-Rex) and recent state-of-the-art (SOTA) methods (e.g., SADE, DirMixE). The results indicate that IDOL outperforms these general invariant learning methods, likely due to the integration of domain-specific knowledge. Moving forward, we plan to include additional invariant representation learning methods in the comparison to enhance the comprehensiveness of the experiments.

---

> > ### Comment · Reviewer_gmF3 · 2025-08-04
> >
> > Dear authors
> >
> > Thank you for the clarification. I am still positive about the work but would maintain my score

---

### Official Review · Reviewer_46TP · 2025-07-21

**Clarity:** 3
**Significance:** 3
**Originality:** 3
**Rating:** 5
**Confidence:** 4

**Summary:**

The paper introduces the deep learning model "IDOL" to reliably estimate parameters of tropical cyclones (TCs) under domain shift (Fig. 1, out-of-distribution predictions). An CNN-based encoder generates a feature vector from timeseries of infrared satellite imagery which serves as an initial estimator of a multi-variate Gaussian distribution (Fig. 3). This distribution is subsequently adjusted by side-information such as the severity level of the TC (Fig. 3 (a)). Further adjustments stem from a knowledge graph correlating the TC's physical properties such as the fullness, energy ratio, etc. to parameters such as wind speed and pressure information (Fig. 3 (b)). All these "identity tokens" model distribution shifts due to shifts in the input imagery and shifts of TC attributes from one TC event to another. The manuscript draws conclusions on the IDOL methodology from experiments with a TC dataset based on "Digital TC" with additional extension resulting in a total of 303 cyclone events - each of which  tracked by approx. 80 timestamps over the Western North Pacific (Tab. 3 & Figs. 5/11/12).

**Questions:**

- The code is not available under https://anonymous.4open.science/r/IDOL-898 as you cited in Question 5, is it?
- How does your approach compare to predicting TCs from state-of-the-art weather foundation models such as Google's GenCast, IBM's PrithviWxC, or Microsoft's Aurora?
- Eq. (2): addtional details on $g$ and $\mathcal{O}_\text{id}$ in supplementary material, please
- Question 10: of course, your work may have significant positive impact on society when accurately predicting cyclone attributes for early warning systems!
- Question 12: Your code will get released under which license?
- Question 13: In l246-l247 you state: "the Physical Dynamic TC datasets (PDTC) we constructed" which I read as you (partially) constructing a new dataset? If not, please clarify and adjust the main text.

**Ethical Concerns:**

["NO or VERY MINOR ethics concerns only"]

**Final Justification:**

The authors did adequately respond to my concerns, cf. my corresponding comment. My decision to accept the work stays.

**Paper Formatting Concerns:**

- l38: "and and" to "and"
- l162: "an" to "a"
- Tab. 1: add colum separators to clearly distinguish "Methods" names such as "MAE", "RMSE", etc.
- l841: broken reference to appendix ("??")

**Quality:**

4

**Strengths And Weaknesses:**

# Strength

- clear presentation of work with carefully crafted figures, well motivated intro
- clean and conistent introduction of math symbols
- straightforward sections well interconnected including links to corresponding appendices

# Weakness

- abstract language of a very complex pipeline may loose some applied ML scientists on reading, suggestion: add specific example with illustration in appendix and add a high-level description in layman's terms to main manuscript
- as acknowledged in Question 7, the model performance experiments do not ship with error bars

---

> ### Author Rebuttal · Authors · 2025-07-29
>
> First of all, we are pleased to see that the reviewers have recognized the clarity of our presentation. We sincerely appreciate all the valuable suggestions, which have greatly helped us to improve the paper. The reviewer’s comments are shown below in italicized font, followed by our responses in regular font. We hope that our responses address the concerns raised and meet with your approval.
>
> **Comment 1**: *abstract language of a very complex pipeline may loose some applied ML scientists on reading, suggestion: add specific example with illustration in appendix and add a high-level description in layman's terms to main manuscript.*
>
> **Response 1**: Thank you for your valuable suggestion. Indeed, tropical cyclone (TC) estimation refers to regressing multiple TC attributes, such as maximum wind speed, pressure, and wind radii, from satellite-based multi-modal data. To better motivate our method and highlight the underlying challenges, Appendix A of the manuscript provides a detailed overview of the different types of distribution shifts present in TC estimation. In response to your suggestion, we will include a specific example with an illustration figure in the revised appendix to better demonstrate the full pipeline in an intuitive and step-by-step manner. Since figures cannot be shown in the rebuttal box, we do not present them here.
>
> **Comment 2**: *as acknowledged in Question 7, the model performance experiments do not ship with error bars.*
>
> **Response 2**: Thank you for your valuable suggestion. We agree that including error bars is important to reflect the model's stability and reliability. We have now supplemented Table 1 in the manuscript with performance metrics including the mean absolute error (MAE), root mean squared error (RMSE), and standard deviation (STD) across multiple runs. The updated results are shown below:
>
> | Method       | Wind MAE | Wind RMSE | Wind STD | Pressure MAE | Pressure RMSE | Pressure STD | Inner-Core MAE | Inner-Core RMSE | Inner-Core STD | Outer-Core MAE | Outer-Core RMSE | Outer-Core STD |
> |--------------|----------|-----------|----------|---------------|----------------|---------------|------------------|-------------------|----------------|------------------|-------------------|----------------|
> | ADT          | 11.20    | 14.20     | –        | 8.00          | 10.20          | –             | –                | –                 | –              | –                | –                 | –              |
> | MTCWA        | –        | –         | –        | –             | –              | –             | 11.70            | 18.20             | –              | 26.40            | 33.00             | –              |
> | STIA         | 10.70    | 14.41     | 9.67     | –             | –              | –             | –                | –                 | –              | –                | –                 | –              |
> | NS           | –        | –         | –        | –             | –              | –             | 10.50            | 15.51             | 11.44          | 26.35            | 36.12             | 24.71          |
> | TC-MTLNet    | 13.82    | 18.06     | 11.62    | 12.00         | 15.46          | 9.79          | –                | –                 | –              | 31.49            | 42.83             | 29.14          |
> | DeepTCNet    | 8.84     | 11.76     | 7.76     | 8.13          | 10.42          | 5.31          | 8.09             | 13.71             | 11.25          | 25.86            | 33.49             | 21.29          |
> | **IDOL**     | **5.84 ± 0.09** | **7.51 ± 0.13** | **4.72 ± 0.12** | **5.75 ± 0.04**     | **7.12 ± 0.05**       | **4.2 ± 0.07**      | **6.21 ± 0.11**         | **12.05 ± 0.2**         | **10.33 ± 0.19**      | **17.17 ± 0.35**         | **23.45 ± 0.44**         | **15.96 ± 0.49**      |
>
> **Note**: Wind, Inner-Core, and Outer-Core refer to estimation tasks of Wind Speed (m/s), Inner-Core Size (nmi), and Outer-Core Size (nmi) errors, respectively.
>
> **In summary, compared with traditional methods (i.e., ADT and MTCWA) and previous state-of-the-art approaches (e.g., DeepTCNet), the proposed IDOL demonstrates superior capability in handling distribution shifts, achieving both the best and most stable estimation performance.**
>
> **Comment 3**: *About available code.*
>
> **Response 3**: As indicated in Question 5 of the checklist, the code is already publicly available at the location we cited, with the correct suffix "IODL-A370". It is possible that the cited text was wrapped in the manuscript and therefore cannot be accessed correctly. Furthermore, we plan to release the code on GitHub under the MIT license in the near future to ensure broader accessibility.
>
> **Comment 4**: *Does your approach compare to predicting TCs from state-of-the-art weather foundation models such as Google's GenCast, IBM's PrithviWxC, or Microsoft's Aurora?*
>
> **Response 4**: Thank you for the valuable review. The proposed IDOL is originally designed for tropical cyclone (TC) estimation. To further demonstrate its transferability and effectiveness under distribution shifts, we adopt TC-Diffuser, the current state-of-the-art (SOTA) model for end-to-end multi-task TC forecasting, as the backbone in a transfer setting. The corresponding results are presented in Table 6 of Appendix C.2 in our manuscript, which demonstrate IDOL’s strong ability to handle distribution shifts, a challenge that is also inevitable in TC forecasting.
>
> Second, regarding TC forecasting, foundation weather models such as Google's GenCast do not adopt an end-to-end architecture for multi-task TC forecasting. Instead, they forecast TC trajectory and intensity by identifying the cyclone center based on Mean Sea Level Pressure (MSLP) in the predicted atmospheric fields from ERA5 datasets. Due to the massive scale of ERA5 datasets, which require substantial time for downloading and processing, we were unable to conduct direct performance comparisons with the mentioned foundation models on the same TC test cases within the limited rebuttal period.
>
> Nevertheless, to further compare the performance of our approach and the foundation weather model, we have supplemented the performance of another publicly available weather foundation model, Pangu (as reported in paper [1]), on comparable TC forecasting tasks, as shown in the following table:
>
> | Model                   | Metric | Traj. 6h | Traj. 12h | Traj. 18h | Traj. 24h | Press. 6h | Press. 12h | Press. 18h | Press. 24h | Wind. 6h | Wind. 12h | Wind. 18h | Wind. 24h |
> |-------------------------|--------|----------|-----------|-----------|-----------|------------|-------------|-------------|-------------|-----------|------------|------------|------------|
> | Pangu                   | MAE    | 42.80    | 44.75     | 50.85     | **65.68**     | 16.00      | 16.50       | 16.70       | 16.90       | –         | –          | –          | –          |
> | TC-Diffuser             | MAE    | 19.39    | 20.83     | 41.82     | 77.41     | 1.24       | 0.84        | 1.85        | 2.81        | 0.75      | 0.43       | 0.93       | 1.38       |
> | **TC-Diffuser w/ IDOL** | MAE    | **18.70**| **19.50** | **40.80** | 74.20 | **1.21**   | **0.81**     | **1.66**     | **2.47**     | **0.73**  | **0.37**    | **0.91**    | **1.31**    |
>
> **Note**: Traj, Press, and Wind refer to forecasting tasks of Trajectory (km), Pressure (hPa), and Wind Speed (m/s) errors, respectively. Additional supplementary experiments comparing our approach with weather foundation models will be included in the revised manuscript.
>
> **In summary, our IDOL is efficient in handling distribution shifts, thus achieving better forecasting performance than the foundation model Pangu.**
> - [1] Zhang, Shiqi, et al. "TC-Diffuser: Bi-Condition Multi-Modal Diffusion for Tropical Cyclone Forecasting." Proceedings of the AAAI Conference on Artificial Intelligence. Vol. 39. No. 1. 2025.
>
> **Comment 5**: *Eq. (2): additional details on $g$ and $\mathcal{O}_\text{id}$ in supplementary material, please.*
>
> **Response 5**: In Eq. (2), the function $g$ refers to the proposed estimation heads based on the Correlation-Aware Attention mechanism, while $\mathcal{O}_\text{id}$ denotes the modules designed to model task identity tokens by leveraging internal invariant physical correlations to regulate feature distributions.
>
> Specifically, $\mathcal{O}_\text{id}$ comprises the proposed Task Dependency Flow Learning and Correlation-Aware Information Bridge. Additional details will be provided in the supplementary material in future revisions.
>
> **Comment 6**: *In l246-l247 you state: "the Physical Dynamic TC datasets (PDTC) we constructed" which I read as you (partially) constructing a new dataset? If not, please clarify and adjust the main text.*
>
> **Response 6**: Yes, the Physical Dynamic TC Datasets (PDTC) is constructed by us. Detailed information and construction procedures of this dataset are provided in Appendix C.1 of the manuscript.
>
> **Comment 7**: *About paper formatting and language.*
>
> **Response 7**: Thank you for your valuable suggestions. We will carefully revise and improve the manuscript based on your comments to further enhance its clarity.

---

> > ### Comment · Reviewer_46TP · 2025-08-04
> > **review of rebuttal**
> >
> > Dear authors: Thank you for your detailed response. A couple comments:
> >
> > - **Response 1**: Wonderful to hear you
> >   > will include a specific example with an illustration figure in the revised appendix to better demonstrate the full pipeline in an intuitive and step-by-step manner
> >
> >    Please make it part of the camera-ready version of your manuscript.
> >
> > - **Response 2**: I am surprised / impressed about the very low error bars. In case you have access: How does it compare to the other methods? I assume the publication of your code and additional experiments by your peers will demonstrate whether these stand the test of time. Personally, I would avoid bold statements such as
> >   > compared with traditional methods (i.e., ADT and MTCWA) and previous state-of-the-art approaches (e.g., DeepTCNet), the proposed IDOL demonstrates superior capability in handling distribution shifts, achieving both the best and most stable estimation performance.
> >
> >   given you estimate the error bars on rerunning your experiments excluding other potential sources of error.
> >
> > - **Response 3**: As it turns out the link followed by clicking the text in Question 5 of the *NeurIPS Paper Checklist* is broken. Copy-pasting the link text works. While scanning the code I noticed a set of Chinese comments. Please convert them to English for the international research community to properly understand your code.
> >
> > - **Response 4**: Thank you for your speedy compilation of additional figures to be included in the supplementary material of the camera-ready version of your manuscript! In the long run, I suggest you compare your efforts to a wide variety of weather foundation models currently published. It'll be educational to see pros and cons of both approaches: foundation models vs. expert models for given downstream tasks.
> >
> > - **Response 5**: Thank you, yes, please elaborate on such math notations and details in the supplementary material in line with **Response 1**.
> >
> > - **Response 6**: Given you introduce a novel dataset, please expand C.1 for the camera-ready manuscript. I believe, the current paragraph **Datasets.** will benefit from some more details regarding your newly curated dataset. For the sake of open-science, please open-source any novel data part of your work.
> >
> > ### Open question
> >
> > - Question 12 (of NeurIPS Paper Checklist): Your code will get released under which license? I highly recommend an open-source, open-science license.

---

> > > ### Author Response · Authors · 2025-08-08
> > >
> > > First of all, we appreciate that the reviewers have taken the time to respond to our rebuttal. The reviewers' comments are shown below in italicized font, followed by our responses in regular font. We hope that our responses address the concerns raised and meet with your approval.
> > >
> > > **Comment 1**: *About additional figures, details and comparison experiments. (For Reviewer’s response 1,4,5)*
> > >
> > > **Response 1**: Thank you for your thoughtful and detailed response. Following the reviewers’ valuable suggestions, we will incorporate the additional figures, mathematical details, and comparison experiments with foundation models in the revised manuscript and supplementary materials. Furthermore, in support of open science, we will release the newly constructed PDTC dataset used in our work as soon as possible.
> > >
> > > **Comment 2**: *I am surprised / impressed about the very low error bars. In case you have access: How does it compare to the other methods? For Reviewer’s response 2 about error bars)*
> > >
> > > **Response 2**: Thank you for your valuable comment. To provide a more comprehensive comparison between our proposed IDOL method and existing approaches, we have conducted additional experiments to supplement the error bars for representative multi-task learning baselines. Due to time constraints, we have added error bars for two TC multi-task estimation methods (TCMTLNet and DeepTCNet), and the updated results are summarized in the table below. In the revised manuscript, we will further include the complete set of error bars for all compared methods.
> > >
> > > | Method       | Wind MAE       | Wind RMSE      | Wind STD       | Pressure MAE   | Pressure RMSE | Pressure STD   | Inner-Core MAE | Inner-Core RMSE | Inner-Core STD | Outer-Core MAE | Outer-Core RMSE | Outer-Core STD |
> > > |--------------|----------------|----------------|----------------|----------------|---------------|----------------|----------------|-----------------|----------------|----------------|-----------------|----------------|
> > > | TC-MTLNet    | 12.72 ± 0.65   | 16.71 ± 0.80   | 10.84 ± 0.48   | 10.46 ± 0.87   | 13.56 ± 1.10  | 8.64 ± 0.68    | –              | –               | –              | 30.94 ± 0.99   | 40.70 ± 1.87   | 26.43 ± 1.91   |
> > > | DeepTCNet    | 8.62 ± 0.15    | 11.40 ± 0.25   | 7.46 ± 0.22    | 8.16 ± 0.12    | 10.43 ± 0.21  | 6.25 ± 0.57    | 8.29 ± 0.35    | 13.90 ± 0.44    | 11.18 ± 0.31   | 25.41 ± 0.55   | 33.84 ± 0.71   | 22.33 ± 0.84   |
> > > | **IDOL**     | **5.84 ± 0.09**| **7.51 ± 0.13**| **4.72 ± 0.12**| **5.75 ± 0.04**| **7.12 ± 0.05**| **4.20 ± 0.07**| **6.21 ± 0.11**| **12.05 ± 0.20**| **10.33 ± 0.19**| **17.17 ± 0.35**| **23.45 ± 0.44**| **15.96 ± 0.49**|
> > >
> > > **Note**: Wind, Inner-Core, and Outer-Core refer to estimation tasks of Wind Speed (m/s), Inner-Core Size (nmi), and Outer-Core Size (nmi) errors, respectively.
> > >
> > > **In summary, compared with previous state-of-the-art approaches, the proposed IDOL demonstrates superior capability in handling distribution shifts, achieving both the best and most stable estimation performance.**
> > >
> > > **Comment 3**: *About available code and datasets. (For Reviewer’s response 5 and open question)*
> > >
> > > **Response 3**: Thank you for the valuable suggestion. We have review the comments in our code and ensure that they are translated into English to facilitate proper understanding by the international research community. Additionally, in compliance with the anonymity requirements of the submission process, the code currently available at the location referenced in Question 5 of the checklist is anonymized. However, we will release our code and dataset on a non-anonymized GitHub repository under an open-source and open-science license (e.g., MIT) to ensure broad accessibility and reproducibility after the review process is completed.

---

### Note · Authors · 2025-08-12

Dear ACs and Reviewers,

We express our gratitude to all reviewers for their insightful comments and valuable suggestions! In the following, we would like to summarize the contributions and responses to the reviews again.

**Contributions and Strengths**

- **Novel Physics-Guided Framework**: The proposed IDOL framework addresses diverse distribution shifts in tropical cyclone (TC) estimation, bridges the gap between invariant learning and meteorological priors, with real-world impact on early warning systems. (Reviewers 46TP, gmF3, 3jY9, SbdX)

- **Well-Designed Architecture**: The method disentangles invariant and variant features guided by physical priors, improving out-of-distribution (OOD) generalization. (Reviewers gmF3, 3jY9)

- **Comprehensive Validation**: Extensive experiments on multiple TC tasks demonstrate IDOL’s robustness to distribution shifts and superior performance over existing methods. (Reviewers gmF3, 3jY9, SbdX).

- **Clear and Well-Motivated Presentation**: The paper is presented with crafted figures, consistent notation, and well-structured sections. (Reviewer 46TP)

**Responses and Discussions**

- **For Reviewer 46TP:**
  • Added error bars showing stability and reliability.
  • Added comparison with Pangu foundation model demonstrating superior forecasting performance.
  • Provided detailed description of dataset construction.

- **For Reviewer gmF3:**
  • Added experiments with approximate priors, confirming their robustness.
  • Clarified motivation via quantified frequency and OOD analysis.
  • Analyzed why the method handles domain shifts.

- **For Reviewer 3jY9:**
  • Added experiments on efficiency showing IDOL’s backbone robustness.
  • Clarified visualization of invariant vs. variant components.
  • Explained efficient integration of new external variables.

- **For Reviewer SbdX:**
  • Supplemented experiments on parameter-sharing effects on model size and multi-task performance.
  • Performed ablation on Holland prior.
  • Clarified invariance ensured by physics-infused design, identity constraints, and experiments.
  • Discussed potential routes for formal proofs.

**Revisions to the Manuscript**
- Will Update experiments with error bars, comparison to weather foundation model, parameter-sharing ablations and prior analyses.
- Committed to release source codes and dataset.

We believe these revisions address all concerns and strengthen our manuscript. Thank you again for your efforts.

---

### Decision · Program_Chairs · 2025-09-17

**Decision:**

Accept (poster)

**Comment:**

This paper introduces IDOL, a physics-guided deep learning framework for tropical cyclone multi-attribute estimation under distribution shifts. The method integrates infrared satellite imagery with physical priors (Holland wind field, dark knowledge correlations) through identity tokens that regulate feature distributions across shared and task-specific components. Experiments on extended Digital TC and a new PDTC dataset demonstrate improved estimation of wind, pressure, and size attributes, and transfer experiments with TC-Diffuser further highlight applicability.

Reviewers converged on the view that the paper is technically solid, novel, and well-motivated, with careful empirical validation and clear positioning in the context of distribution shifts in meteorology. Strengths emphasized include the creative use of physics priors, elegant task dependency modeling, and consistent empirical gains. The main concerns raised were the lack of formal invariance guarantees, questions about the necessity of priors and ablations, and evaluation breadth relative to stronger baselines. The rebuttal was detailed: the authors provided ablations on the role of the Holland model, sensitivity analysis on shared vs task-specific tokens, error bars, and clarified dataset construction. This successfully resolved several reviewer doubts (leading to one upgraded rating), though some reviewers maintained their scores, citing the absence of formal proofs and limited broader validation.

Overall, the consensus after rebuttal and discussion is that the contributions are meaningful, methodologically sound, and supported by substantial evidence. While not flawless, the paper advances physics-informed invariant learning for climate applications, and its ideas are likely to stimulate follow-up work. I recommend acceptance.